psychology

transparency, reproducibility, meta-research, social sciences, open science

**Author for correspondence:**
Tom E. Hardwicke
e-mail: tom.hardwicke@charite.de

# An empirical assessment of transparency and reproducibility-related research practices in the social sciences (2014–2017)

Tom E. Hardwicke[1,2], Joshua D. Wallach[3,4,5], Mallory C. Kidwell[6], Theiss Bendixen[7], Sophia Crüwell[1] and John P. A. Ioannidis[1,2,8]

[1]Meta-Research Innovation Center Berlin (METRIC-B), QUEST Center for Transforming Biomedical Research, Berlin Institute of Health, Charité – Universitätsmedizin Berlin, Anna-Louisa-Karsch-Str.2, 10178 Berlin, Germany
[2]Meta-Research Innovation Center at Stanford (METRICS), Stanford University, Stanford, CA
[3]Department of Environmental Health Sciences, Yale School of Public Health, New Haven, CT
[4]Center for Outcomes Research and Evaluation, Yale-New Haven Hospital, New Haven, CT
[5]Collaboration for Research Integrity and Transparency, Yale School of Medicine, New Haven, CT
[6]Department of Psychology, University of Utah, Salt Lake City, UT
[7]Department of the Study of Religion, Aarhus University, Aarhus, Denmark
[8]Departments of Medicine, of Health Research and Policy, of Biomedical Data Science, and of Statistics, Stanford University, Stanford, CA

TEH, 0000-0001-9485-4952

Serious concerns about research quality have catalysed a number of reform initiatives intended to improve transparency and reproducibility and thus facilitate self-correction, increase efficiency and enhance research credibility. Meta-research has evaluated the merits of some individual initiatives; however, this may not capture broader trends reflecting the cumulative contribution of these efforts. In this study, we manually examined a random sample of 250 articles in order to estimate the prevalence of a range of transparency and reproducibility-related indicators in the social sciences literature published between 2014 and 2017. Few articles indicated availability of materials (16/151, 11% [95% confidence interval, 7% to 16%]), protocols (0/156, 0% [0% to 1%]), raw data (11/156, 7% [2% to 13%]) or analysis scripts (2/156, 1% [0% to 3%]), and no studies were pre-registered (0/156, 0% [0% to 1%]). Some articles explicitly disclosed funding sources (or lack of; 74/236, 31% [25% to 37%]) and some declared no conflicts of interest (36/236, 15% [11% to 20%]). Replication studies were rare (2/156, 1% [0% to 3%]). Few studies were included in

evidence synthesis via systematic review (17/151, 11% [7% to 16%]) or meta-analysis (2/151, 1% [0% to 3%]). Less than half the articles were publicly available (101/250, 40% [34% to 47%]). Minimal adoption of transparency and reproducibility-related research practices could be undermining the credibility and efficiency of social science research. The present study establishes a baseline that can be revisited in the future to assess progress.

# 1. Introduction

Transparency and reproducibility are core scientific principles and their pursuit can be important for improving efficiency [1], facilitating self-correction [2] and enhancing the credibility of the published literature [3]. Although journal articles are often considered the principal commodity of the scientific ecosystem [4], they usually only provide an incomplete summary of a research project. A rich array of research resources such as (ideally pre-registered) protocols (i.e. hypotheses, methods and analysis plans), materials, raw data and analysis scripts provide the most detailed documentation of a study, and having access to them can support independent verification, replication, evidence synthesis and further discovery [5]. For example, access to raw data enables assessment of computational reproducibility [6–10] and analytic robustness [11], more sophisticated individual participant-level meta-analyses [12], and creative merging of datasets or use of novel analytic techniques [13]. However, although researchers appear to endorse the values of transparency and reproducibility in principle [14,15], they are routinely neglected in practice [16–19].

Concerns about low transparency and poor reproducibility [20–24] have catalysed an array of reform initiatives from researchers, journals, publishers, funders and universities. In some cases, it has been possible to isolate these interventions and evaluate their impact through meta-research (research on research) [25,26]. For example, journal data sharing policies have been associated with an increased quantity of publicly available datasets [6,27,28]. However, such assessments may not capture broader trends reflecting the cumulative contribution of disparate initiatives and growing awareness among researchers about the importance of transparency and reproducibility [15]. It is, therefore, valuable to zoom out and take a bird's eye view of the published literature. Building on similar studies in the biomedical domain [17,18], we manually examined a random sample of 250 articles in order to estimate the prevalence of a range of transparency and reproducibility-related indicators in the social sciences literature published between 2014 and 2017.

# 2. Methods

The study protocol was pre-registered on 3 July 2018 and is available at https://osf.io/u5bk9/. An amended version of the protocol was registered on 14th August 2019 and is available at https://osf.io/j5zsh/. All deviations from the original protocol are explicitly acknowledged in the main text. All data exclusions and measures conducted during this study are reported.

## 2.1. Design

This was an observational study with a cross-sectional design. Measured variables are shown in table 1.

## 2.2. Sample

### 2.2.1. Sampling frame

We identified a random sample of 250 articles published in the social sciences between January 2014 and April 2017. Specifically, the sample was drawn from a database based on Scopus-indexed content that classifies academic documents according to one of 12 broad fields of science using a model of the disciplinary structure of the scientific literature [29]. The model grouped documents into 91 726 clusters using citation information and 14 342 of these clusters were assigned to the social sciences category. For the time period of interest (2014–2017), the 14 342 social science clusters contained 648 063 documents with an article type of 'article' or 'review' (as determined by Scopus). However, not all articles included in these social science clusters may be strongly related to social sciences. Therefore, to further ensure the face validity of articles included in the sample, we additionally limited the documents to those that had

**Table 1.** Measured variables. The variables measured for an individual article depended on the study design classification. Additionally, for articles that were not available (the full text could not be retrieved), no other variables were measured. The exact operational definitions and procedures for data extraction/coding are available in the structured form here: https://osf.io/v4f59/.

|  | applicable study designs |
| --- | --- |
| **articles** | |
| accessibility and retrieval method (can the article be accessed, is there a public version or is paywall access required?) | all |
| **protocols** | |
| availability statement (is availability, or lack of, explicitly declared?) | empirical studies[a], commentaries and meta-analyses |
| content (what aspects of the study are included in the protocol?) | |
| **materials** | |
| availability statement (is availability, or lack of, explicitly declared?) | empirical studies[a] |
| retrieval method (e.g. upon request or via online repository) | |
| accessibility (can the materials be accessed?) | |
| **raw data** | |
| availability statement (is availability, or lack of, explicitly declared?) | empirical studies[a], commentaries and meta-analyses |
| retrieval method (e.g. upon request or via online repository) | |
| accessibility (can the data be accessed?) | |
| content (has all relevant data been shared?) | |
| documentation (are the data understandable?) | |
| **analysis scripts** | |
| availability statement (is availability, or lack of, explicitly declared?) | empirical studies[a], commentaries and meta-analyses |
| retrieval method (e.g. upon request or via online repository) | |
| accessibility (can the scripts be accessed?) | |
| **pre-registration** | |
| availability statement (is availability, or lack of, explicitly declared?) | empirical studies[a], commentaries and meta-analyses |
| retrieval method (which registry was used?) | |
| accessibility (can the pre-registration be accessed?) | |
| content (what was pre-registered?) | |
| **funding** | |
| disclosure statement (are funding sources, or lack of, explicitly declared?) | all |
| conflicts of interest | |
| disclosure statement (are conflicts of interest, or lack of, explicitly declared?) | all |
| **replication** | |
| statement (does the article claim to report a replication?) | all |
| citation history (has the article been cited by a study that claims to be a replication?) | empirical studies[a] |
| **evidence synthesis** | |
| meta-analysis citation history (has the article been cited by, and included in the evidence-synthesis component of, a meta-analysis) | empirical studies[a] |

**Table 1.** (*Continued.*)

| | applicable study designs |
|---|---|
| systematic review citation history (has the article been cited by, and included in the evidence-synthesis component of, a systematic review) | empirical studies[a] |
| **article characteristics** | |
| subject area, year of publication, study design, country of origin (based on corresponding author's affiliation), human/animal subjects, 2016 journal impact factor (according to Thomson Reuters Journal Citation Reports) | all |

[a]'Empirical studies' encompasses the following study design classifications: field studies, laboratory studies, surveys, case studies, multiple types, clinical trials and 'other' designs.

an All Science Journal Classification (ASJC) code related to the social sciences, specifically 'Economics, Econometrics and Finance' (ECON), 'Psychology' (PSYCH), 'Business, Management and Accounting' (BUS) and 'Social Sciences' (SOC). Note that this means the sample would not capture social science articles published in multidisciplinary journals (e.g. *Nature, Science, PNAS, RSOS*) and/or journals that belong mainly to other disciplines. The number of documents in the database broken by ASJC code was as follows: BUS, 105 447; ECON, 92 348; PSYCH, 75 353; SOC, 324 618.

Based on our judgement of what sample size would be sufficiently large enough to be informative but also realistically feasible for manual data extraction, we decided to examine 250 articles. Random sampling was performed by using a random number generator to shuffle the order of the articles and selecting the top N articles required. Following insightful comments from a peer-reviewer, we detected a (now corrected) error in the original sampling procedure. Full details of the error and the steps taken to correct it are available at https://osf.io/7anx6/. The final sample of 250 articles represents a random sample of the 485 460 English-language articles available in the database [29] that were classified into one of the social sciences clusters, also had an ASJC code specifically related to the social sciences (BUS, ECON, PSYCH or SOC), and were published between January 2014 and April 2017.

## 2.3. Procedure

A structured form (https://osf.io/v4f59/) was created based on previous investigations in the biomedical domain [17,18] and used to guide data extraction and coding for a range of indicators related to transparency and reproducibility. The exact indicators measured depended on study design (table 1). We pilot tested the data extraction procedures using 15 articles that were not included in the final sample.

Between 50 and 71 articles were randomly assigned to each of four investigators (JDW, MCK, TB and SC) who performed initial extraction and coding. All articles were second coded by a fifth investigator (TEH) and any coding differences were resolved through discussion. Journal impact factors and citations histories were obtained by a single investigator only. Data collection took place between 7 May 2018 and 10 October 2019.

We attempted to access each article by searching with the Open Access Button (https://openaccessbutton.org/), Google (https://www.google.com/), and two of five university libraries (Stanford University, Yale University, University of Utah, Aarhus University, Charité – Universitätsmedizin Berlin) depending on the institutional affiliation of the assigned coders.

## 3. Results

Values in square brackets are 95% confidence intervals based on the Sison-Glaz method for multinomial proportions [30].

## 3.1. Sample characteristics

Accessible articles originated from journals with a wide range of journal impact factors (median 1.30, range 0.04–16.79; based on 2016 journal impact factor). For 122 articles, no journal impact factor was available. Other sample characteristics are displayed in table 2.

**Table 2.** Sample characteristics for the 236 accessible articles.

| | *n* |
|---|---|
| **subject area** | |
| education | 34 |
| geography, planning and development | 18 |
| sociology and political science | 16 |
| cultural studies | 14 |
| economics and econometrics | 14 |
| business and international management | 12 |
| finance | 10 |
| 44 other social sciences subject-areas[a] (accounting for less than 10 per area) | 118 |
| **year of publication** | |
| 2014 | 63 |
| 2015 | 84 |
| 2016 | 69 |
| 2017 | 20 |
| **study design** | |
| no empirical data | 80 |
| field study | 73 |
| survey | 39 |
| multiple types | 22 |
| case study | 9 |
| laboratory study | 7 |
| commentary with analysis | 3 |
| meta-analysis | 2 |
| other | 1 |
| **country of origin** | |
| USA | 76 |
| UK | 25 |
| Australia | 17 |
| Germany | 13 |
| 47 other countries[b] (accounting for less than 10 per country) | 105 |
| **subjects** | |
| human | 105 |
| animal | 0 |
| neither human nor animal subjects involved | 131 |

[a]For all subject areas, see https://osf.io/7fm96/.
[b]For all countries, see https://osf.io/kg7j5/.

## 3.2. Article availability

Among the 250 articles, 101 (40% [34% to 47%]) had a publicly available version (figure 1*a*) whereas 135 (54% [48% to 61%]) were only accessible to us through a paywall. Of those paywalled articles, 34 (25%) had received public funding (this may be an underestimate because 67% of the paywalled articles did not provide a funding statement); 14 (6% [0% to 12%]) additional articles were not available to our team, highlighting that even researchers with broad academic access privileges cannot reach portions of the scientific literature.

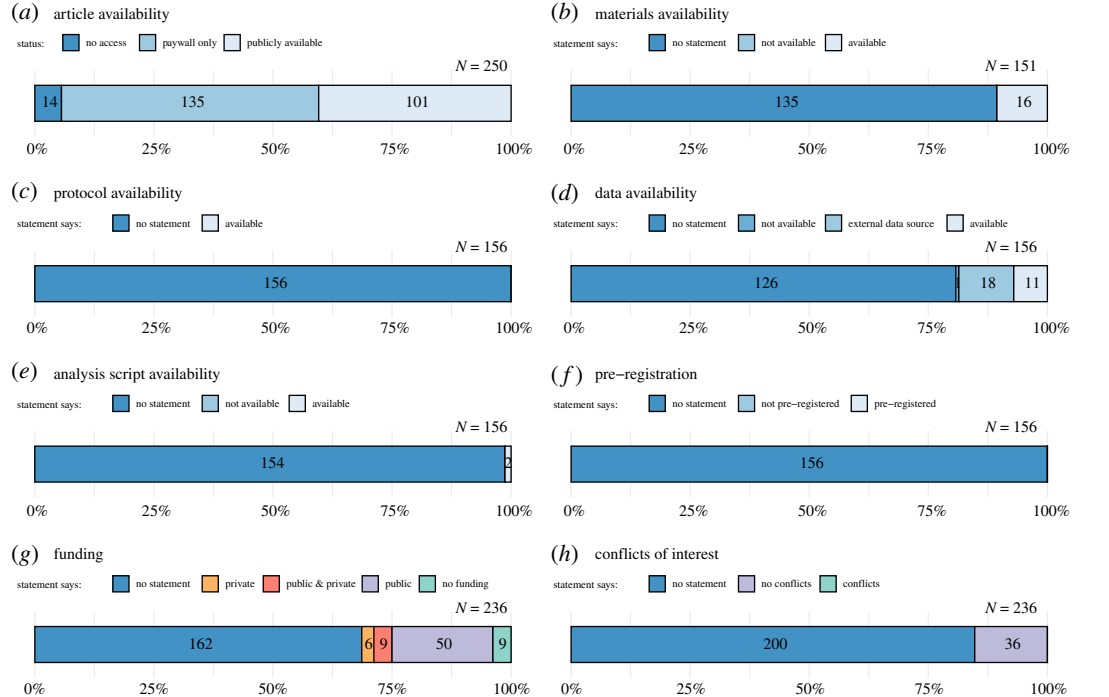

**Figure 1.** Assessment of transparency and reproducibility-related research practices in the social sciences. Numbers inside bars indicate raw counts. 'N' refers to total articles assessed for a given indicator (which was contingent on study design, table 1).

## 3.3. Materials and protocol availability

Research protocols (documentation of the research methodology and analysis plan) and original research materials (such as survey instruments, stimuli or software) provide a detailed account of how a study was performed. The vast majority of articles we examined did not contain a materials availability statement (figure 1b) and none contained a protocol availability statement (figure 1c). Of the 16 of 151 (11% [7% to 16%]) articles where materials were reportedly available, two links to journal-hosted electronic supplementary material were broken and 1 was only 'available upon request' from authors. The remaining 13 sets of materials were available in the article itself ($n = 10$), via journal-hosted electronic supplementary material ($n = 2$), or in another paper ($n = 1$).

## 3.4. Data availability

Raw data are the core evidence that underlies scientific claims. However, the vast majority of the 156 relevant articles we examined did not contain a data availability statement (figure 1d). Eighteen articles (12% [6% to 18%]) mentioned using an external data source,[1] but did not provide detailed instructions on how to obtain the specific data that was used. A further 11 (7% [2% to 13%]) datasets were reportedly available; however, eight of these were in fact not accessible due to non-functioning links to personal/institutional web pages ($n = 4$), a non-functioning link to journal electronic supplementary material ($n = 1$), only being 'available upon request' from authors ($n = 2$; note that we did not attempt to contact them), or involving a file that could not be opened because commercial software was required ($n = 1$). Of the three accessible datasets (two available via electronic supplementary material and one available via a personal/institutional web page) two were both complete and clearly documented and one was neither complete nor clearly documented.

## 3.5. Analysis script availability

Analysis scripts provide detailed step-by-step descriptions of performed analyses, often in the form of computer code (e.g. R, Python or Matlab) or syntax (SPSS, Stata, SAS). Two of 156 (1% [0% to 3%])

---

[1]This particular classification was created *post hoc* and not included in the pre-registered protocol.

articles reported that analysis scripts were available (figure 1e) via journal-hosted electronic supplementary material. One of these was not accessible due to a non-functioning link.

## 3.6. Pre-registration

Pre-registration refers to the archiving of a read-only, time-stamped study protocol in a public repository (such as the Open Science Framework, https://osf.io/) prior to study commencement [31]. None of the articles specified that the study was pre-registered (figure 1f), highlighting that this concept still has fledgling status in the social sciences.

## 3.7. Conflicts of interest and funding statements

Statements about potential conflicts of interest and funding sources enable researchers to disclose pertinent factors that could introduce bias: 162 of 236 (69% [63% to 75%]) articles did not include a funding statement (figure 1g), 200 of 236 (85% [81% to 89%]) articles did not include a conflicts of interest statement (figure 1h), and 36 of 236 (15% [11% to 20%]) articles reported that there were no conflicting interests and no articles reported that there were conflicting interests. Of the articles that reported their funding sources, 59 had received public funding.

## 3.8. Replication and evidence synthesis

Replication studies repeat the methods of previous studies in order to systematically gather evidence on a given research question. Evidence gathered across multiple relevant studies can be formally collated and synthesized through systematic reviews and meta-analyses. Only 2 of the 156 (1% [0% to 3%]) articles we examined explicitly self-identified as a replication study. No articles were cited by another article reporting to be a replication. Seventeen (11% [7% to 16%]) articles were included in evidence synthesis in a systematic review. Two (1% [0% to 3%]) had been included in evidence synthesis in a meta-analysis (an additional 1 was cited but excluded). Overall, articles associated with empirical data were infrequently cited (median = 1, min = 0, max = 31).

# 4. Discussion

Our empirical assessment of a random sample of articles published between 2014 and 2017 suggests a serious neglect of transparency and reproducibility in the social sciences. Most research resources, such as materials, protocols, raw data and analysis scripts, were not explicitly available, no studies were pre-registered, disclosure of funding sources and conflicts of interest was modest, and replication or evidence synthesis via meta-analysis or systematic review was rare.

Poor transparency can have very real costs. For example, the Reproducibility Project in Cancer Biology—a major effort to replicate 50 high-impact cancer biology papers—recently had to abandon 32 replication attempts partly because pertinent methodological information about the original studies was not available [32]. Conversely, transparency and sharing can have tangible benefits. For example, a paper entitled 'Growth in a Time of Debt', which influenced government austerity policies around the world, was found to contain a serious analytic error by a student examining formulae and data in the original spreadsheet [33]. Social science research often addresses topics that are highly pertinent to policy makers, and policy decisions based on flawed research can have substantial economic, social and individual costs. To ensure that a credible evidence base is available to policy makers, it is imperative that social science research is held to high-quality standards [22].

The finding that less than half of the articles in our sample were publicly available is broadly consistent with a large-scale automated analysis which suggested that 55% of the scientific literature published in 2015 was not open access [34]. This suggests that critical stakeholders are potentially being deprived of relevant scientific information. Six per cent of the articles in our sample could not even be retrieved through some of the most comprehensive university library catalogues.

The low availability of materials and protocols that we observed is consistent with transdisciplinary investigations indicating that most articles lack availability statements for these resources [16–18]. Not being able to view original research materials and protocols precludes comprehensive evaluation of a study [3]. Furthermore, inability to obtain materials and protocols can disrupt or even prevent efforts to conduct high-fidelity replication attempts which are vital for both verification and the systematic

accumulation of knowledge [2]. We did not evaluate whether all relevant research materials had been made available or whether protocols were sufficiently described in the articles themselves.

Assessments in other scientific disciplines have also shown that raw data are typically not available [10,16–18] including for some of the most influential articles [35]. Requesting data directly from authors typically yields only a modestly successful response [10,35–37]. Even data that have been made available can be poorly organized and documented, and do not guarantee the analytic reproducibility of the reported outcomes [6–10,38]. The inability to access the raw data that underlines scientific claims seriously limits verification and reuse, potentially wasting resources by impeding discovery and compromising the supposedly self-correcting nature of the scientific endeavour [1–3].

Sharing of analysis scripts seems extremely rare in many scientific fields [6,7,18]. Analysis scripts are important because the verbal description of the original analyses provided in a published paper can be ambiguous, incomplete, or even incorrect, undermining efforts to independently establish computational reproducibility [6–9] and precluding comprehensive peer scrutiny [3].

We found no studies had been pre-registered. Because pre-registration occurs prior to study commencement, there will be some lag before any changes in the uptake of pre-registration are detectable in the published literature. Additionally, pre-registration may be less pertinent when analyses of pre-existing data are intended to be entirely exploratory and no pre-conceived protocol really exists. Various modes of study pre-registration are gaining attention in several research communities, but still capture only a small proportion of total research output [31]. At the time of writing, the Open Science Framework contains over 41 000 pre-registrations (https://osf.io/registries/discover). Pre-registration is intended to facilitate the demarcation of exploratory (unplanned) and confirmatory (planned) analyses, and potentially mitigates or enables detection of questionable research practices, such as opaque 'hypothesizing after the results are known' [39], selective reporting [19,40,41] or '$p$-hacking' [31,42,43]. When studies are not pre-registered, these potential advantages are lost.

Disclosure statements related to potential conflicts of interest and funding are also often neglected in the biomedical sciences [17,18], but the situation seems worse in the social sciences. Articles that do not include these disclosure statements cannot be evaluated in full context. It is possible that many authors neglect to include these statements because they believe there is nothing relevant to disclose. Nevertheless, a lack of funding or potential conflicts is still important to explicitly acknowledge because the absence of a statement is ambiguous.

We assessed article citation histories in order to gauge how often they had been cited overall and cited by replication studies, meta-analyses or systematic reviews specifically. It should be noted that our sample pertained to recently published studies and it may take some time before studies that build upon the original articles are themselves published. Nevertheless, these findings are generally consistent with evidence suggesting that replication studies are rarely conducted or reported and individual studies only infrequently contribute to the cumulative body of research evidence through inclusion in systematic reviews and meta-analyses [17,18,44].

Our study has several important limitations. Firstly, no database has perfect coverage of all journals in every field, but Scopus coverage is extensive. Google Scholar may have broader coverage than Scopus, but it is less transparent than Scopus about what journals and sources of information are included. Secondly, we relied solely on published information. If we had contacted authors, it is likely that we would have been able to obtain additional information. However, requests to authors for resources like raw data typically have modest yield and the modal outcome is to not respond [35–37,45]. Thirdly, as with previous similar investigations [17,18], we did not deeply assess the quality of information and resources that were shared. Given the low levels of transparency, such quality checks would not have been especially informative. However, it is important to bear in mind that transparency alone is rarely sufficient. For example, just because data are shared does not necessarily imply that they are reusable in practice [6]. Fourthly, achieving transparency is not always straightforward when there are overriding legal, ethical or practical concerns [46]. It is possible that a lack of transparency (in particular, a lack of data sharing) is well justified in some cases. However, no such justifications were stated in the articles we examined. It has been argued that when complete transparency is not possible, the reasons should be explicitly declared in any relevant scientific publications [47]. Finally, although our sample can be used to estimate the prevalence of the measured indicators broadly in the social sciences, those estimates may not necessarily generalize to specific contexts, for example, specific subfields or articles published in specific journals. It is known, for example, that specific journal policies can be associated with increases in data and materials availability [6,27,28].

Minimal adoption of transparency and reproducibility-related research practices impoverishes the scientific ecosystem [1], disrupts self-correction activities [2] and could undermine public trust in science [3]. A cascade of reform initiatives have been launched in an effort to improve transparency [21,22], and close monitoring and evaluation of specific reform initiatives will help to identify efforts that are the most successful and highlight areas for improvement [25,26]. The present study establishes a baseline estimate for the prevalence of transparency and reproducibility-related indicators in the social sciences that can be revisited in the future to assess progress.

Data accessibility. All data (https://osf.io/u9fw8/), materials (https://osf.io/z9qtr/) and analysis scripts (https://osf.io/sbrez/) related to this study are publicly available. To facilitate reproducibility this manuscript was written by interleaving regular prose and analysis code and is available in a Code Ocean container (https://doi.org/10.24433/CO.2749769.v4) which re-creates the software environment in which the original analyses were performed.

Authors' contributions. T.E.H., J.D.W. and J.P.A.I. designed the study. T.E.H., J.D.W., M.C.K., T.B. and S.C. conducted the data collection. T.E.H. performed the data analysis. T.E.H., J.D.W. and J.P.A.I. wrote the manuscript. All the authors gave their final approval for publication.

Competing interests. In the past 36 months, J.D.W. received research support outside of the scope of this project through the Collaboration for Research Integrity and Transparency from the Laura and John Arnold Foundation, and through the Center for Excellence in Regulatory Science and Innovation (CERSI) at Yale University and the Mayo Clinic (U01FD005938). All other authors declare no competing interests.

Funding. The authors received no specific funding for this work. The Meta-Research Innovation Center at Stanford (METRICS) is supported by a grant from the Laura and John Arnold Foundation. The Meta-Research Innovation Center Berlin (METRIC-B) is supported by a grant from the Einstein Foundation and Stiftung Charité.

Acknowledgements. We thank Kevin Boyack for assistance obtaining the sample of articles. We are grateful to an anonymous peer-reviewer whose comments led us to identify an error (now corrected) in our initial sampling procedure.

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
