## [Reviewer comments · Royal Society Open Science]

Review History

RSOS-190806.R0 (Original submission)

Review form: Reviewer 1 (Nate Breznau)

Is the manuscript scientifically sound in its present form?

No

Are the interpretations and conclusions justified by the results?

No

Is the language acceptable?

No

Is it clear how to access all supporting data?

Not Applicable

Do you have any ethical concerns with this paper?

No

Have you any concerns about statistical analyses in this paper?

No

Recommendation?

Reject

Comments to the Author(s)

There seems to be some serious confusion. The paper takes stock of practices in the social sciences and sets itself as a benchmark against which to evaluate future progress in the social sciences. Thus, it is of absolutely no relevance to the sciences this journal covers only to "life sciences, physical sciences, mathematics, engineering and computer science". I emailed the editor and asked about this and he confirmed that, "The Royal Society, our publisher, is the UK's national scientific academy, but it does not generally tackle the clinical or social sciences nor the humanities." So this paper should have been desk rejected, and when I pointed this out I did not get a response. So I am sorry to have to reject it here but it absolutely does not fit in a journal that specifically does not publish social science research. The fact that the paper was not desk rejected speaks poorly about the editorial process of what otherwise seems like a journal that could make a good contribution to open sciences, at least in its limited 'hard' sciences scope.

Review form: Reviewer 2

Is the manuscript scientifically sound in its present form?

Yes

Are the interpretations and conclusions justified by the results?

No

Is the language acceptable?

Yes

Is it clear how to access all supporting data?

Yes

Do you have any ethical concerns with this paper?

No

Have you any concerns about statistical analyses in this paper?

Yes

Recommendation?

Reject

Comments to the Author(s)

The "Subject" and "Subject Category" of this study are reported as ": psychology < BIOLOGY" and Psychology and cognitive neuroscience" although as already the title reveals this study is about "transparency and reproducibility-related research practices in the _social sciences_"

Looking at the database used it surprises that it includes numerous studies from journals that do not or certainly not primarily cover social sciences. Methodologically focused journals like Canadian Journal of Statistics <https://www.scimagojr.com/journalsearch.php?q=28893&tip=sid> Computational Statistics <https://www.scimagojr.com/journalsearch.php?q=28930&tip=sid>

Journal of Applied Probability <https://www.scimagojr.com/journalsearch.php?q=23838&tip=sid>
 Communications in Statistics Part B: Simulation and Computation
<https://www.scimagojr.com/journalsearch.php?q=23526&tip=sid>
 Computational and Applied Mathematics
<https://www.scimagojr.com/journalsearch.php?q=5000153703&tip=sid>
 Linear Algebra and Its Applications
<https://www.scimagojr.com/journalsearch.php?q=24475&tip=sid>
 Mathematical Inequalities & Applications
<https://www.scimagojr.com/journalsearch.php?q=24572&tip=sid>
 may still be acceptable. As the above links to the publicly available Scimago Journal Rank information of Scopus show, Scopus categorizes the first three of these journals also under “decision sciences”. The topics of the studies in several cases however are difficult to categorize as social sciences.

In many other cases it is completely unclear how the studies could end up in a sample of articles from the social sciences. There are studies from journals like
 Plastics Engineering (the doi is missing, it is 10.1002/j.1941-9635.2015.tb01322.x and the Scopus link shows there is no connection whatsoever to the social sciences
<https://www.scimagojr.com/journalsearch.php?q=14353&tip=sid> and even the database provided categorizes this as “Materials Chemistry”),
 CHEMICAL ENGINEERING TRANSACTIONS (Scopus and the provided database classify this journal under chemical engineering, not a subfield of social sciences:
<https://www.scimagojr.com/journalsearch.php?q=19600161818&tip=sid>),
 BMJ Open (<https://www.scimagojr.com/journalsearch.php?q=19800188003&tip=sid> the study topic arguably falls under social sciences but the journal is classified as Medicine as one would assume from the abbreviation (British Medical Journal...)),
 Journal of Engineering for Gas Turbines and Power
<https://www.scimagojr.com/journalsearch.php?q=20962&tip=sid>,
 Frontiers of Information Technology and Electronic Engineering
<https://www.scimagojr.com/journalsearch.php?q=21100409130&tip=sid>
 European Journal of Paediatric Dentistry
<https://www.scimagojr.com/journalsearch.php?q=25027&tip=sid>
 Carpathian Journal of Earth and Environmental Sciences
<https://www.scimagojr.com/journalsearch.php?q=15900154727&tip=sid>
 Wounds UK <https://www.scimagojr.com/journalsearch.php?q=4500151403&tip=sid>
 Canadian Family Physician <https://www.scimagojr.com/journalsearch.php?q=110256&tip=sid>
 Indian Journal of Science and Technology
<https://www.scimagojr.com/journalsearch.php?q=21100201522&tip=sid>
 Physical review. E <https://www.scimagojr.com/journalsearch.php?q=21100855841&tip=sid> (in this case surprisingly at least the subject of the study can be described as social science even though Scopus categorizes the journal under mathematics)
 Journal of Uncertain Systems
<https://www.scimagojr.com/journalsearch.php?q=19900191975&tip=sid>
 Journal of Intellectual Disability Research
<https://www.scimagojr.com/journalsearch.php?q=16726&tip=sid>
 Fire Rescue Magazine <https://www.scimagojr.com/journalsearch.php?q=5000156910&tip=sid>
 Journal of Vocational Rehabilitation
<https://www.scimagojr.com/journalsearch.php?q=29285&tip=sid>
 Journal of Clinical Urology
<https://www.scimagojr.com/journalsearch.php?q=21100235629&tip=sid>
 European Journal of Philosophy
<https://www.scimagojr.com/journalsearch.php?q=5600155103&tip=sid>
 Archives of Physical Medicine and Rehabilitation
<https://www.scimagojr.com/journalsearch.php?q=16270&tip=sid>
 Journal of Nutrition and Health
<https://www.scimagojr.com/journalsearch.php?q=21100259127&tip=sid>

Zhongguo Jixie Gongcheng/China Mechanical Engineering
<https://www.scimagojr.com/journalsearch.php?q=22181&tip=sid>
 Computers in Industry <https://www.scimagojr.com/journalsearch.php?q=19080&tip=sid>
 Journal of Musicology <https://www.scimagojr.com/journalsearch.php?q=14000155925&tip=sid>
 Pacific Historical Review <https://www.scimagojr.com/journalsearch.php?q=23676&tip=sid>
 (study topic arguably social science but journal categorized as “arts and humanities – history”)
 Statistics in Medicine <https://www.scimagojr.com/journalsearch.php?q=20086&tip=sid>
 twice The Chaucer Review (correctly categorized as “literature” in the database)
<https://www.scimagojr.com/journalsearch.php?q=13243&tip=sid>
 International Journal of Developmental Neuroscience
<https://www.scimagojr.com/journalsearch.php?q=16147&tip=sid>
 Lecture Notes in Computer Science
<https://www.scimagojr.com/journalsearch.php?q=25674&tip=sid>
 twice Revista Facultad de Ingenieria
<https://www.scimagojr.com/journalsearch.php?q=12400154740&tip=sid>
 (in one case a social science topic but the journal is not categorized as social science)
 Biogeosciences <https://www.scimagojr.com/journalsearch.php?q=130037&tip=sid>
 Currents in Pharmacy Teaching and Learning
<https://www.scimagojr.com/journalsearch.php?q=19500157042&tip=sid>
 Diabetes Primary Care <https://www.scimagojr.com/journalsearch.php?q=5200152617&tip=sid>
 SMT Surface Mount Technology Magazine
<https://www.scimagojr.com/journalsearch.php?q=27175&tip=sid>

If these studies were indeed found using a Scopus database this may be useful to illustrate that this Scopus database is not very useful in identifying social science research. The data presented is certainly not useful to make assessments about the state of transparency in the social sciences. To be honest I am not even sure if this may just be a test whether reviewers actually look at underlying data of a study if they have the chance because based on the data that is made available this analysis does not make any sense at all.

Unfortunately, I had already used some of my time to start a referee report under the assumption that this was a serious analysis. I leave it in the current state as given what I saw in the database I see no value in writing a full report and strongly recommend a rejection of this study.

In the article it is claimed that the study was pre-registered. In the pre-registration report, it is stated: “Of the 215 eligible articles, we have randomly selected 15 to be used for piloting purposes, leaving 200 eligible articles in the main sample”. This statement shows that the pre-registration already included the result of the sampling process. It is furthermore not explained how the randomization was conducted, and the piloting purposes are not motivated or explained in any way.

It is not defined what “raw data” exactly means in the study. Given the big differences in study designs it would have been necessary to define clearly in advance what kind of raw data is expected in which case (and why). The policy of a number of journals that raw data need to be made available to other researchers on request is not mentioned.

It is not explained how it was checked if a study had been replicated, been part of a systematic review or a meta-analysis. Which citation databases were investigated? Scopus again?

What exactly is meant with “2016 journal impact factor”? 2-year-SSCI or Scimago Journal Rank? The latter would have been more appropriate to report given that the studies were selected from Scopus, which includes much more journals than the SSCI. It seems more likely that SSCI was used given that it is reported many of the analyzed studies were published in journals for which no impact factor was available. To assess how big the sample drawn for the study is compared to the whole distribution of social science studies in the SCOPUS database it should be stated how many studies were classified as social science in Scopus in 2014-2017 altogether. Without this number the reported “95% confidence intervals based on the Sison-Glaz method for multinomial proportions” (that I am not familiar with) are not convincing. Altogether it seems questionable how such a small sample can help to draw conclusions on research in the social sciences in

general. It is not investigated how transparency measures vary across journals of different rank, and with such a small sample size this should be impossible. Why did the authors then not use available data from many previous studies that analyzed the many aspects mentioned instead? One additional detail that was neglected in the analysis: Focusing on journals categorized as “social science” in Scopus excludes all studies from the social sciences that were published in journals like Nature, Science, the Proceedings of the National Academy of Sciences or Royal Society Open Science that cover all sciences. At least this should be mentioned and the importance of such studies in comparison to all social science studies should be assessed. While the number will be relatively slow, the impact of such studies will on average be much higher than that of most others because they were deemed important enough to be published in such general interest journals.

The variable “country of origin” is not defined: Of the data? Of the authors or their affiliations? If the latter, how were cases with more than one author treated?

The “wide range of journal impact factors (median 1.33, range 0.25-16.79; based on 2016 journal impact factor)” should be compared to the whole distribution of thousands of journals. Studies from how many journals out of a distribution of how many total social science Scopus journals were investigated? How does the distribution of journal impact factors in the sample compare to the distribution of impact factors of all Scopus social science journals? For just some thousand journals these informative descriptive statistics are not difficult to obtain. The same holds for values of the whole distribution for subject areas and years of publication.

The judgment “articles associated with empirical data were rarely cited at all (median = 1, min = 0, max = 31)” does not seem justified given the reported median journal impact factor of 1.33 – how is 1 low as compared to 1.33? From which databases were citing articles identified and when? Given that the sample includes studies published in 2017 and it seems likely that the cited journal impact factors refer to the last two years, it is not surprising that the empirical studies had not yet been cited so often by at the latest May 2019.

In the conclusion it is pointed out that “social science research often addresses topics that are highly pertinent to policy makers and policy decisions based on flawed research can have substantial economic, social, and individual costs.” This is illustrated as follows: “a paper entitled “Growth in a Time of Debt” which influenced government austerity policies around the world was found to contain a serious analytic error by a student examining formulas and data in the original spreadsheet.” While both statements are true, to analyze the impact of the lack of transparency on policies it is necessary to investigate papers that really have an impact and not a random sample of studies covered in the Scopus database. As is well known, most academic studies are hardly ever cited even in academia and certainly do not have any impact beyond. The study “Growth in a Time of Debt” was published in 2010 in a journal with one of the highest impact factors in the social sciences and had already been cited more than a thousand times by 2014. To argue about the impact of lack of transparency in social science one needs to analyze the transparency of impactful social science studies, which are typically published in more cited journals, which tend to have much stronger transparency rules than the average journal. In addition, to assess the impact on policy one may not focus on citations in academia but in policy papers or in the non-scientific media. Similarly, when discussing the state of open access it should be taken into account that in social sciences like economics none of the most highly ranked journals are published open access or even give the option to publish papers open access, so open access has so far been a phenomenon mostly for low prestige research. It would be interesting to see how this varies across sub-disciplines, time and journal ranks, but with a sample of 200 randomly selected studies over three years it will be impossible to make a real assessment of any of these aspects.

In the discussion of the advantages of preregistration an important point is left out: In social sciences where data is often readily available there is no way to know whether a “pre-registration” was actually registered before or after an analysis (with potential p-hacking and HARKing) was done.

The discussion of possible reasons for lack of transparency includes the following statement: “It is possible that a lack of transparency is well justified for some of the articles we have examined. However, such justifications were not made explicit.” Numerous social science journals publish reports about data availability, including cases of sensitive or proprietary data, and there are

guidelines how to inform about such restrictions. If none such case was in the sample this just reveals that the chosen sample size was too small to allow an informative assessment of the state of transparency in the social sciences.

Charitè is spelled with a wrong accent (should be Charité).

As a last comment, the authors heavily cite previous studies from just one social science, psychology, many of whose sub-disciplines share more characteristics with life sciences than with other social sciences. The one exception is a study from economics that is used as an example, which does not suit its purpose well. It would be beneficial to the study to draw on the existing literature on transparency in the social sciences from fields like political sciences, sociology, and economics.

Review form: Reviewer 3 (Edward Miguel)

Is the manuscript scientifically sound in its present form?

Yes

Are the interpretations and conclusions justified by the results?

Yes

Is the language acceptable?

Yes

Is it clear how to access all supporting data?

Yes

Do you have any ethical concerns with this paper?

No

Have you any concerns about statistical analyses in this paper?

No

Recommendation?

Accept with minor revision (please list in comments)

Comments to the Author(s)

“An empirical assessment of transparency and reproducibility-related practices in the social sciences (2014-2017)”

There is a lot to like about this manuscript, and I am broadly supportive of publication, provided that the authors are able make some minor revisions.

The manuscript provides an important snapshot of the state of transparency practices in the social sciences today. Figure 1 is destined to be widely commented on, and taught in courses. The fact that the analysis is pre-registered is a major strength. Overall, I find the analysis credible and the findings innovative.

I would suggest a few possible edits to the authors.

(1) The authors should comment more on the small sample size of articles included in the analysis (N=250 initially, which falls to 198 in the final analysis due mainly to an English language exclusion criterion). Given the important of this topic, and the relatively low cost of classifying articles, why such a small sample? A larger sample would produce a more definitive statistical analysis. This is an important limitation of the current analysis that should be flagged for the reader.

(2) The authors (rightfully) point out that, given their recent publication dates, few of the articles could reasonably already be included in meta-analyses or systematic reviews. Given this, I would

remove presentation of these results from the main figure and text, and instead relegate them to the appendix. This would save space in the main article.

(3) There is too little detail on the nature of the social science articles that are considered in the analysis. If the authors seek to inform a social science research community audience, they will need to do a better job of characterizing the sample of journals, scholars, articles, and topics that they focus on here. One interpretation is that the paper was written by medical researchers with limited knowledge of the "terrain" they are operating in here, in the social sciences. The authors inform us that very few of the articles, only 17 out of 198, are in Economics, Political Science, or Sociology, three of the largest, best cited and most influential social science fields for public policy. This is a disappointment, since it feels (to me) like this sample of articles may not be very representative of influential social science research. Economics and Political Science are also social science fields with some of the most visible open science "movements". Some specific questions. What fields are the N=125 articles that are in "other sub-areas" actually in? The fact that around 40% of articles in the sample also have "no empirical data" leaves me wondering what types of studies these are. There are many possibilities - theoretical / conceptual work, ethnographies, historical work (without original data), critical theory - but which of these are in the sample being considered here? Overall, this came across to me as the most important limitation of the article. I do think that having a larger sample size of articles, and perhaps stratifying the random sampling of articles in a more systematic way - perhaps oversampling articles in journals with higher impact factors, or making sure you have at least some minimal representation of major social science fields, i.e., more than just 8 articles in Sociology and Political Science combined! - would go a long way in future work along these lines.

Decision letter (RSOS-190806.R0)

03-Jul-2019

Dear Dr Hardwicke,

The editors assigned to your paper ("An empirical assessment of transparency and reproducibility-related research practices in the social sciences (2014-2017)") have now received comments from reviewers. We would like you to revise your paper in accordance with the referee and Associate Editor suggestions which can be found below (not including confidential reports to the Editor). Please note this decision does not guarantee eventual acceptance.

Please submit a copy of your revised paper before 26-Jul-2019. Please note that the revision deadline will expire at 00.00am on this date. If we do not hear from you within this time then it will be assumed that the paper has been withdrawn. In exceptional circumstances, extensions may be possible if agreed with the Editorial Office in advance. In this case requesting such extension may be appropriate, as the associate editor has suggested very extensive reworking of the paper. We do not allow multiple rounds of revision so we urge you to make every effort to fully address all of the comments at this stage. If deemed necessary by the Editors, your manuscript will be sent back to one or more of the original reviewers for assessment. If the original reviewers are not available, we may invite new reviewers.

When submitting your revised manuscript, you must respond to the comments made by the

referees and upload a file "Response to Referees" in "Section 6 - File Upload". Please use this to document how you have responded to the comments, and the adjustments you have made. In order to expedite the processing of the revised manuscript, please be as specific as possible in your response.

- Data accessibility

If you wish to submit your supporting data or code to Dryad (<http://datadryad.org/>), or modify your current submission to dryad, please use the following link:
<http://datadryad.org/submit?journalID=RSOS&manu=RSOS-190806>

- Competing interests

- Authors' contributions

- Acknowledgements

- Funding statement

on behalf of Professor Chris Chambers (Associate Editor) and Essi Viding (Subject Editor)
openscience@royalsociety.org

Associate Editor's comments (Professor Chris Chambers):

Associate Editor: 1

Comments to the Author:

Three expert reviewers have now appraised this manuscript. The assessments are mixed overall. Reviewer 1 recommends rejection on the grounds that the article falls out of scope for RSOS. The issue of scope is an editorial judgment, and in sending the manuscript for in-depth review, the journal has already decided that the focus on reproducibility and open research practices places it within scope, similar to previous articles published in this journal. Therefore, as much as the journal is grateful for the assessment of Reviewer 1, the authors not need respond to this review. Reviewer 2 is also very negative for different reasons, chiefly that the inclusion criteria are unclear (or poorly chosen), and that a variety of methodological details are either unspecified or unjustified. Finally, Reviewer 3 is more positive but raises concerns with the selection of articles and sample size, echoing the concerns of Reviewer 2. Overall, the reviews are too negative to proceed with publication, and the manuscript falls very close to the line for outright rejection. However given the positive assessment of Reviewer 3, and the overall importance of this question, I recommend a comprehensive Major Revision. This is likely to require additional data collection and analysis, quite possibly in deviation from the preregistered protocol, and it may be that the authors will prefer to voluntarily withdraw their submission rather than embark on the substantial extra work that is likely to be needed to satisfy Reviewers 2 and 3.

Comments to Author:

Reviewers' Comments to Author:

Reviewer: 1

Comments to the Author(s)

There seems to be some serious confusion. The paper takes stock of practices in the social sciences and sets itself as a benchmark against which to evaluate future progress in the social sciences. Thus, it is of absolutely no relevance to the sciences this journal covers only to "life sciences, physical sciences, mathematics, engineering and computer science". I emailed the editor and asked about this and he confirmed that, "The Royal Society, our publisher, is the UK's national scientific academy, but it does not generally tackle the clinical or social sciences nor the humanities." So this paper should have been desk rejected, and when I pointed this out I did not get a response. So I am sorry to have to reject it here but it absolutely does not fit in a journal that specifically does not publish social science research. The fact that the paper was not desk rejected speaks poorly about the editorial process of what otherwise seems like a journal that could make a good contribution to open sciences, at least in its limited 'hard' sciences scope.

Reviewer: 2

Comments to the Author(s)

The “Subject” and “Subject Category” of this study are reported as “: psychology < BIOLOGY” and Psychology and cognitive neuroscience” although as already the title reveals this study is about “transparency and reproducibility-related research practices in the _social sciences_”

Looking at the database used it surprises that it includes numerous studies from journals that do not or certainly not primarily cover social sciences. Methodologically focused journals like Canadian Journal of Statistics <https://www.scimagojr.com/journalsearch.php?q=28893&tip=sid> Computational Statistics <https://www.scimagojr.com/journalsearch.php?q=28930&tip=sid> Journal of Applied Probability <https://www.scimagojr.com/journalsearch.php?q=23838&tip=sid> Communications in Statistics Part B: Simulation and Computation <https://www.scimagojr.com/journalsearch.php?q=23526&tip=sid> Computational and Applied Mathematics <https://www.scimagojr.com/journalsearch.php?q=5000153703&tip=sid> Linear Algebra and Its Applications <https://www.scimagojr.com/journalsearch.php?q=24475&tip=sid> Mathematical Inequalities & Applications <https://www.scimagojr.com/journalsearch.php?q=24572&tip=sid> may still be acceptable. As the above links to the publicly available Scimago Journal Rank information of Scopus show, Scopus categorizes the first three of these journals also under “decision sciences”. The topics of the studies in several cases however are difficult to categorize as social sciences.

In many other cases it is completely unclear how the studies could end up in a sample of articles from the social sciences. There are studies from journals like Plastics Engineering (the doi is missing, it is 10.1002/j.1941-9635.2015.tb01322.x and the Scopus link shows there is no connection whatsoever to the social sciences <https://www.scimagojr.com/journalsearch.php?q=14353&tip=sid> and even the database provided categorizes this as “Materials Chemistry”), CHEMICAL ENGINEERING TRANSACTIONS (Scopus and the provided database classify this journal under chemical engineering, not a subfield of social sciences: <https://www.scimagojr.com/journalsearch.php?q=19600161818&tip=sid>), BMJ Open (<https://www.scimagojr.com/journalsearch.php?q=19800188003&tip=sid> the study topic arguably falls under social sciences but the journal is classified as Medicine as one would assume from the abbreviation (British Medical Journal...)), Journal of Engineering for Gas Turbines and Power <https://www.scimagojr.com/journalsearch.php?q=20962&tip=sid>, Frontiers of Information Technology and Electronic Engineering <https://www.scimagojr.com/journalsearch.php?q=21100409130&tip=sid> European Journal of Paediatric Dentistry <https://www.scimagojr.com/journalsearch.php?q=25027&tip=sid> Carpathian Journal of Earth and Environmental Sciences <https://www.scimagojr.com/journalsearch.php?q=15900154727&tip=sid> Wounds UK <https://www.scimagojr.com/journalsearch.php?q=4500151403&tip=sid> Canadian Family Physician <https://www.scimagojr.com/journalsearch.php?q=110256&tip=sid> Indian Journal of Science and Technology <https://www.scimagojr.com/journalsearch.php?q=21100201522&tip=sid> Physical review. E <https://www.scimagojr.com/journalsearch.php?q=21100855841&tip=sid> (in this case surprisingly at least the subject of the study can be described as social science even though Scopus categorizes the journal under mathematics) Journal of Uncertain Systems <https://www.scimagojr.com/journalsearch.php?q=19900191975&tip=sid> Journal of Intellectual Disability Research <https://www.scimagojr.com/journalsearch.php?q=16726&tip=sid>

Fire Rescue Magazine <https://www.scimagojr.com/journalsearch.php?q=5000156910&tip=sid>
 Journal of Vocational Rehabilitation
<https://www.scimagojr.com/journalsearch.php?q=29285&tip=sid>
 Journal of Clinical Urology
<https://www.scimagojr.com/journalsearch.php?q=21100235629&tip=sid>
 European Journal of Philosophy
<https://www.scimagojr.com/journalsearch.php?q=5600155103&tip=sid>
 Archives of Physical Medicine and Rehabilitation
<https://www.scimagojr.com/journalsearch.php?q=16270&tip=sid>
 Journal of Nutrition and Health
<https://www.scimagojr.com/journalsearch.php?q=21100259127&tip=sid>
 Zhongguo Jixie Gongcheng/China Mechanical Engineering
<https://www.scimagojr.com/journalsearch.php?q=22181&tip=sid>
 Computers in Industry <https://www.scimagojr.com/journalsearch.php?q=19080&tip=sid>
 Journal of Musicology <https://www.scimagojr.com/journalsearch.php?q=14000155925&tip=sid>
 Pacific Historical Review <https://www.scimagojr.com/journalsearch.php?q=23676&tip=sid>
 (study topic arguably social science but journal categorized as “arts and humanities – history”)
 Statistics in Medicine <https://www.scimagojr.com/journalsearch.php?q=20086&tip=sid>
 twice The Chaucer Review (correctly categorized as “literature” in the database)
<https://www.scimagojr.com/journalsearch.php?q=13243&tip=sid>
 International Journal of Developmental Neuroscience
<https://www.scimagojr.com/journalsearch.php?q=16147&tip=sid>
 Lecture Notes in Computer Science
<https://www.scimagojr.com/journalsearch.php?q=25674&tip=sid>
 twice Revista Facultad de Ingenieria
<https://www.scimagojr.com/journalsearch.php?q=12400154740&tip=sid>
 (in one case a social science topic but the journal is not categorized as social science)
 Biogeosciences <https://www.scimagojr.com/journalsearch.php?q=130037&tip=sid>
 Currents in Pharmacy Teaching and Learning
<https://www.scimagojr.com/journalsearch.php?q=19500157042&tip=sid>
 Diabetes Primary Care <https://www.scimagojr.com/journalsearch.php?q=5200152617&tip=sid>
 SMT Surface Mount Technology Magazine
<https://www.scimagojr.com/journalsearch.php?q=27175&tip=sid>

If these studies were indeed found using a Scopus database this may be useful to illustrate that this Scopus database is not very useful in identifying social science research. The data presented is certainly not useful to make assessments about the state of transparency in the social sciences. To be honest I am not even sure if this may just be a test whether reviewers actually look at underlying data of a study if they have the chance because based on the data that is made available this analysis does not make any sense at all.

Unfortunately, I had already used some of my time to start a referee report under the assumption that this was a serious analysis. I leave it in the current state as given what I saw in the database I see no value in writing a full report and strongly recommend a rejection of this study.

In the article it is claimed that the study was pre-registered. In the pre-registration report, it is stated: “Of the 215 eligible articles, we have randomly selected 15 to be used for piloting purposes, leaving 200 eligible articles in the main sample”. This statement shows that the pre-registration already included the result of the sampling process. It is furthermore not explained how the randomization was conducted, and the piloting purposes are not motivated or explained in any way.

It is not defined what “raw data” exactly means in the study. Given the big differences in study designs it would have been necessary to define clearly in advance what kind of raw data is expected in which case (and why). The policy of a number of journals that raw data need to be made available to other researchers on request is not mentioned.

It is not explained how it was checked if a study had been replicated, been part of a systematic review or a meta-analysis. Which citation databases were investigated? Scopus again? What exactly is meant with “2016 journal impact factor”? 2-year-SSCI or Scimago Journal Rank? The latter would have been more appropriate to report given that the studies were selected from Scopus, which includes much more journals than the SSCI. It seems more likely that SSCI was used given that it is reported many of the analyzed studies were published in journals for which no impact factor was available. To assess how big the sample drawn for the study is compared to the whole distribution of social science studies in the SCOPUS database it should be stated how many studies were classified as social science in Scopus in 2014-2017 altogether. Without this number the reported “95% confidence intervals based on the Sison-Glaz method for multinomial proportions” (that I am not familiar with) are not convincing. Altogether it seems questionable how such a small sample can help to draw conclusions on research in the social sciences in general. It is not investigated how transparency measures vary across journals of different rank, and with such a small sample size this should be impossible. Why did the authors then not use available data from many previous studies that analyzed the many aspects mentioned instead? One additional detail that was neglected in the analysis: Focusing on journals categorized as “social science” in Scopus excludes all studies from the social sciences that were published in journals like Nature, Science, the Proceedings of the National Academy of Sciences or Royal Society Open Science that cover all sciences. At least this should be mentioned and the importance of such studies in comparison to all social science studies should be assessed. While the number will be relatively slow, the impact of such studies will on average be much higher than that of most others because they were deemed important enough to be published in such general interest journals.

The variable “country of origin” is not defined: Of the data? Of the authors or their affiliations? If the latter, how were cases with more than one author treated?

The “wide range of journal impact factors (median 1.33, range 0.25-16.79; based on 2016 journal impact factor)” should be compared to the whole distribution of thousands of journals. Studies from how many journals out of a distribution of how many total social science Scopus journals were investigated? How does the distribution of journal impact factors in the sample compare to the distribution of impact factors of all Scopus social science journals? For just some thousand journals these informative descriptive statistics are not difficult to obtain. The same holds for values of the whole distribution for subject areas and years of publication.

The judgment “articles associated with empirical data were rarely cited at all (median = 1, min = 0, max = 31)” does not seem justified given the reported median journal impact factor of 1.33 – how is 1 low as compared to 1.33? From which databases were citing articles identified and when? Given that the sample includes studies published in 2017 and it seems likely that the cited journal impact factors refer to the last two years, it is not surprising that the empirical studies had not yet been cited so often by at the latest May 2019.

In the conclusion it is pointed out that “social science research often addresses topics that are highly pertinent to policy makers and policy decisions based on flawed research can have substantial economic, social, and individual costs.” This is illustrated as follows: “a paper entitled “Growth in a Time of Debt” which influenced government austerity policies around the world was found to contain a serious analytic error by a student examining formulas and data in the original spreadsheet.” While both statements are true, to analyze the impact of the lack of transparency on policies it is necessary to investigate papers that really have an impact and not a random sample of studies covered in the Scopus database. As is well known, most academic studies are hardly ever cited even in academia and certainly do not have any impact beyond. The study “Growth in a Time of Debt” was published in 2010 in a journal with one of the highest impact factors in the social sciences and had already been cited more than a thousand times by 2014. To argue about the impact of lack of transparency in social science one needs to analyze the transparency of impactful social science studies, which are typically published in more cited journals, which tend to have much stronger transparency rules than the average journal. In addition, to assess the impact on policy one may not focus on citations in academia but in policy papers or in the non-scientific media. Similarly, when discussing the state of open access it should be taken into account that in social sciences like economics none of the most highly ranked journals are published open access or even give the option to publish papers open access, so open

access has so far been a phenomenon mostly for low prestige research. It would be interesting to see how this varies across sub-disciplines, time and journal ranks, but with a sample of 200 randomly selected studies over three years it will be impossible to make a real assessment of any of these aspects.

In the discussion of the advantages of preregistration an important point is left out: In social sciences where data is often readily available there is no way to know whether a “pre-registration” was actually registered before or after an analysis (with potential p-hacking and HARKing) was done.

The discussion of possible reasons for lack of transparency includes the following statement: “It is possible that a lack of transparency is well justified for some of the articles we have examined. However, such justifications were not made explicit.” Numerous social science journals publish reports about data availability, including cases of sensitive or proprietary data, and there are guidelines how to inform about such restrictions. If none such case was in the sample this just reveals that the chosen sample size was too small to allow an informative assessment of the state of transparency in the social sciences.

Charitè is spelled with a wrong accent (should be Charité).

As a last comment, the authors heavily cite previous studies from just one social science, psychology, many of whose sub-disciplines share more characteristics with life sciences than with other social sciences. The one exception is a study from economics that is used as an example, which does not suit its purpose well. It would be beneficial to the study to draw on the existing literature on transparency in the social sciences from fields like political sciences, sociology, and economics.

Reviewer: 3

Comments to the Author(s)

“An empirical assessment of transparency and reproducibility-related practices in the social sciences (2014-2017)”

There is a lot to like about this manuscript, and I am broadly supportive of publication, provided that the authors are able make some minor revisions.

The manuscript provides an important snapshot of the state of transparency practices in the social sciences today. Figure 1 is destined to be widely commented on, and taught in courses. The fact that the analysis is pre-registered is a major strength. Overall, I find the analysis credible and the findings innovative.

I would suggest a few possible edits to the authors.

- (1) The authors should comment more on the small sample size of articles included in the analysis (N=250 initially, which falls to 198 in the final analysis due mainly to an English language exclusion criterion). Given the important of this topic, and the relatively low cost of classifying articles, why such a small sample? A larger sample would produce a more definitive statistical analysis. This is an important limitation of the current analysis that should be flagged for the reader.
- (2) The authors (rightfully) point out that, given their recent publication dates, few of the articles could reasonably already be included in meta-analyses or systematic reviews. Given this, I would remove presentation of these results from the main figure and text, and instead relegate them to the appendix. This would save space in the main article.
- (3) There is too little detail on the nature of the social science articles that are considered in the analysis. If the authors seek to inform a social science research community audience, they will need to do a better job of characterizing the sample of journals, scholars, articles, and topics that they focus on here. One interpretation is that the paper was written by medical researchers with limited knowledge of the “terrain” they are operating in here, in the social sciences. The authors inform us that very few of the articles, only 17 out of 198, are in Economics, Political Science, or Sociology, three of the largest, best cited and most influential social science fields for public policy. This is a disappointment, since it feels (to me) like this sample of articles may not be very representative of influential social science research. Economics and Political Science are also social science fields with some of the most visible open science “movements”. Some specific

questions. What fields are the N=125 articles that are in “other sub-areas” actually in? The fact that around 40% of articles in the sample also have “no empirical data” leaves me wondering what types of studies these are. There are many possibilities – theoretical / conceptual work, ethnographies, historical work (without original data), critical theory – but which of these are in the sample being considered here? Overall, this came across to me as the most important limitation of the article. I do think that having a larger sample size of articles, and perhaps stratifying the random sampling of articles in a more systematic way – perhaps oversampling articles in journals with higher impact factors, or making sure you have at least some minimal representation of major social science fields, i.e., more than just 8 articles in Sociology and Political Science combined! – would go a long way in future work along these lines.

Author's Response to Decision Letter for (RSOS-190806.R0)

See Appendix A.

RSOS-190806.R1 (Revision)

Review form: Reviewer 2

Is the manuscript scientifically sound in its present form?

No

Are the interpretations and conclusions justified by the results?

No

Is the language acceptable?

Yes

Do you have any ethical concerns with this paper?

No

Have you any concerns about statistical analyses in this paper?

Yes

Recommendation?

Reject

Comments to the Author(s)

Subject:psychology < BIOLOGY and Subject Category:Psychology and cognitive neuroscience still do not fit. If the journal chooses to accept social science contributions, appropriate categories should be offered to submitting authors.

I find a bit confusing that there are two different files for the text, one apparently with changes from the original manuscript in red color and the other one the revised manuscript but I could not find this explained anywhere.

In my first round review I commented that a large part of the studies analyzed were taken from journals that do not primarily cover social sciences, that were classified as covering different fields by the Scopus database that the authors used and that a number of the studies had topics that did not resemble social science research. The authors reduce this comment to “The reviewer highlights a number of journals in which articles in the sample were published and says that those journals do not primarily cover social sciences research.” They claim this is “not relevant [seriously] because the articles were not defined as social sciences research by the journal they were published in, but by a citation clustering method”. I cannot help but ask the authors: If you need a reviewer to see that these engineering, literature, philosophy, medicine or chemistry studies have nothing whatsoever to do with social sciences, how come you feel qualified to do research about the social sciences? I find embarrassing to read you even contacted another researcher to confirm that you had included many of the studies in your sample based on a sample of his in which he apparently had not even classified these studies as social science as you claimed. How come you could not figure this out yourself? How come that the origin of such an error can remain “unknown” in a pre-registered, supposedly well documented study on transparency and reproducibility?

It would certainly also be interesting to analyze what kind of social science research is published in journals that do not primarily have a social science focus. This aspect should however have been made clear to the reader from the start. Additional aspects should then be considered as for example how do they differ from standard social science contributions – there will be reasons why authors address a different audience. Given that the authors themselves choose for other parts of their study standard approaches looking at Scopus classifications by type of research or Thomson Reuters impact factors I find the mixed sampling method surprising and it is not clear to me why this approach was chosen. I cannot follow the authors’ calculation for the sample size they regard as sufficient and cannot help but find the size of 200 studies for all social sciences together and for the number of aspects investigated alarmingly small.

The data provided under the link <https://osf.io/3d5um/> includes only 250 studies. It is not clear at first sight which ones were among the 108 from the previous version of the study, why in a study on transparency and reproducibility the 65 that had originally been miscategorized and the 27 that were later disregarded because of the added journal classification criterion as well as those that were disregarded as not in the English language are hidden now, and which ones were the 15 “used for piloting purposes”. Why is the full sample of 600 studies not shown, especially by researchers who write they use “raw data” as synonymous to “data”?

Looking at the new data I find it questionable to consider studies on chemistry and nursing (education) or literary history in an analysis of transparency in the social sciences. But studies that are already categorized in your own data as philosophy and arts and humanities?

I see that one has to limit one’s research in some ways, just as an aside I’d like to note that the Scopus citation database that was used here is very incomplete even compared to readily available other data as for example that of google scholar.

The choice to use the corresponding author’s location as “country of origin” of a study does not convince me. Many studies nowadays are written by multinational teams and the corresponding author’s location is not necessarily informative about where the most important parts of the research were performed.

It is unclear to me how the word “rarely” could ever be meant in an “absolute” rather than a relative sense. I would say I have rarely read a study on social sciences that investigated philosophy and theology studies. In geological terms once in a human lifetime would usually not be considered a rare event. Analogously, one citation per study is not “rare” in bibliometric terms where the median and mode citation count is 1 or even 0 in most social sciences unless the sample of studies is restricted in particular ways.

I continue to see the example of the Growth in a Time of Debt article as grossly misleading about the importance of the present study as it is not at all representative for a sample of randomly selected studies from all social (and some not so social) sciences. This should be made clear and not just be left to the reader’s interpretation.

I maintain the point that when discussing the benefits of pre-registration also the shortcoming should be noted that when data is readily available as is most often the case in the social sciences it is impossible to know whether an analysis was actually done before or after a “pre”-

registration. This does not apply in cases where data collection requires large funds that can usually only be acquired with a previously registered plan as is often the case in randomized controlled trials in, for example, medicine or development economics. Here pre-registration gives studies much more credibility but this cannot be generalized to the social sciences.

I also maintain my point that if the authors did not find a single case in which data was not made available because for legitimate reasons sensitive or proprietary data was used and this is explained by the journal this illustrates that the chosen sample was too small to allow an informative assessment of the state of transparency in the social sciences. At the very least informative parts of the literature should be cited and summarized that inform about the many journals that have specific rules about such data that cannot easily be shared.

How can you agree that your literature review should not exclusively contain papers from psychology but also from social sciences like political sciences, sociology, and economics, and then add references ONLY to economics papers??? Many aspects you are investigating have been investigated for even tiny subfields of the different disciplines of the social sciences and the sample sizes are often much bigger even for such studies. You just don't do a proper literature review and then try to convince your readers that you can contribute anything with just 200 articles that you chose with a randomization process that first made you analyze papers that totally fall out of the scope of your study?

It is my understanding that a reviewer's job is to help select between not so good and good contributions for publication and give advice on how to improve the latter. Not to help authors who are able to do this themselves to improve their manuscripts to make them acceptable for publication.

My judgment is already made from the comments above and I refuse to spend more time reading the manuscript again as I regard this inadequate use of a peer's voluntarily contributed time. Given that also reviewer one already refused to look at the manuscript again under the circumstances I would advise the authors to prepare their research better before submission to academic journals. They are certainly able to do so as at least one of them has already published high quality inspiring research in the past. I am looking forward to reading more of that in the future. If you need any help or advice you know how to reach me. I am sorry I cannot be of more help at this point.

Decision letter (RSOS-190806.R1)

02-Jan-2020

Dear Dr Hardwicke:

Manuscript ID RSOS-190806.R1 entitled "An empirical assessment of transparency and reproducibility-related research practices in the social sciences (2014-2017)" which you submitted to Royal Society Open Science, has been reviewed. The comments of the reviewer(s) are included at the bottom of this letter.

Please submit a copy of your revised paper before 25-Jan-2020. Please note that the revision deadline will expire at 00.00am on this date. If we do not hear from you within this time then it will be assumed that the paper has been withdrawn. In exceptional circumstances, extensions may be possible if agreed with the Editorial Office in advance. We do not generally allow multiple rounds of revision so we urge you to make every effort to fully address all of the comments at this stage. The reviewer has a very negative view of the paper, but the AE is of the opinion that you have worked to improve the paper and should be allowed one more opportunity to provide a final response/revision. The AE and myself will discuss the response and revision, should you decide to submit such, but cannot guarantee publication.

- Ethics statement

- Data accessibility

- Competing interests

- Authors' contributions

- Acknowledgements

- Funding statement

Kind regards,

Andrew Dunn

on behalf of Professor Chris Chambers (Associate Editor) and Essi Viding (Subject Editor)

Associate Editor Comments to Author (Professor Chris Chambers):

One reviewer (Reviewer 2) was available to assess the revised submission. The reviewer's assessment remains very negative. However, as AE, in my reading of the revised submission and the re-review, I find little justification for so strong a recommendation given the multiple improvements the manuscript (including the correction of the uncovered extraction error). Nevertheless, the reviewer remains deeply unconvinced, and I would like to see the authors' response to this reviewer's concerns before rendering a final decision. This decision will be reached through editorial assessment without further in-depth review.

Reviewer comments to Author:

Reviewer: 2

Comments to the Author(s)

Subject:psychology < BIOLOGY and Subject Category:Psychology and cognitive neuroscience still do not fit. If the journal chooses to accept social science contributions, appropriate categories should be offered to submitting authors.

I find a bit confusing that there are two different files for the text, one apparently with changes from the original manuscript in red color and the other one the revised manuscript but I could not find this explained anywhere.

In my first round review I commented that a large part of the studies analyzed were taken from journals that do not primarily cover social sciences, that were classified as covering different fields by the Scopus database that the authors used and that a number of the studies had topics that did not resemble social science research. The authors reduce this comment to "The reviewer highlights a number of journals in which articles in the sample were published and says that those journals do not primarily cover social sciences research." They claim this is "not relevant [seriously] because the articles were not defined as social sciences research by the journal they were published in, but by a citation clustering method". I cannot help but ask the authors: If you need a reviewer to see that these engineering, literature, philosophy, medicine or chemistry studies have nothing whatsoever to do with social sciences, how come you feel qualified to do research about the social sciences? I find embarrassing to read you even contacted another researcher to confirm that you had included many of the studies in your sample based on a

sample of his in which he apparently had not even classified these studies as social science as you claimed. How come you could not figure this out yourself? How come that the origin of such an error can remain “unknown” in a pre-registered, supposedly well documented study on transparency and reproducibility?

It would certainly also be interesting to analyze what kind of social science research is published in journals that do not primarily have a social science focus. This aspect should however have been made clear to the reader from the start. Additional aspects should then be considered as for example how do they differ from standard social science contributions – there will be reasons why authors address a different audience. Given that the authors themselves choose for other parts of their study standard approaches looking at Scopus classifications by type of research or Thomson Reuters impact factors I find the mixed sampling method surprising and it is not clear to me why this approach was chosen. I cannot follow the authors’ calculation for the sample size they regard as sufficient and cannot help but find the size of 200 studies for all social sciences together and for the number of aspects investigated alarmingly small.

The data provided under the link <https://osf.io/3d5um/> includes only 250 studies. It is not clear at first sight which ones were among the 108 from the previous version of the study, why in a study on transparency and reproducibility the 65 that had originally been miscategorized and the 27 that were later disregarded because of the added journal classification criterion as well as those that were disregarded as not in the English language are hidden now, and which ones were the 15 “used for piloting purposes”. Why is the full sample of 600 studies not shown, especially by researchers who write they use “raw data” as synonymous to “data”?

Looking at the new data I find it questionable to consider studies on chemistry and nursing (education) or literary history in an analysis of transparency in the social sciences. But studies that are already categorized in your own data as philosophy and arts and humanities?

I see that one has to limit one’s research in some ways, just as an aside I’d like to note that the Scopus citation database that was used here is very incomplete even compared to readily available other data as for example that of google scholar.

The choice to use the corresponding author’s location as “country of origin” of a study does not convince me. Many studies nowadays are written by multinational teams and the corresponding author’s location is not necessarily informative about where the most important parts of the research were performed.

It is unclear to me how the word “rarely” could ever be meant in an “absolute” rather than a relative sense. I would say I have rarely read a study on social sciences that investigated philosophy and theology studies. In geological terms once in a human lifetime would usually not be considered a rare event. Analogously, one citation per study is not “rare” in bibliometric terms where the median and mode citation count is 1 or even 0 in most social sciences unless the sample of studies is restricted in particular ways.

I continue to see the example of the Growth in a Time of Debt article as grossly misleading about the importance of the present study as it is not at all representative for a sample of randomly selected studies from all social (and some not so social) sciences. This should be made clear and not just be left to the reader’s interpretation.

I maintain the point that when discussing the benefits of pre-registration also the shortcoming should be noted that when data is readily available as is most often the case in the social sciences it is impossible to know whether an analysis was actually done before or after a “pre”-registration. This does not apply in cases where data collection requires large funds that can usually only be acquired with a previously registered plan as is often the case in randomized controlled trials in, for example, medicine or development economics. Here pre-registration gives studies much more credibility but this cannot be generalized to the social sciences.

I also maintain my point that if the authors did not find a single case in which data was not made available because for legitimate reasons sensitive or proprietary data was used and this is explained by the journal this illustrates that the chosen sample was too small to allow an informative assessment of the state of transparency in the social sciences. At the very least informative parts of the literature should be cited and summarized that inform about the many journals that have specific rules about such data that cannot easily be shared.

How can you agree that your literature review should not exclusively contain papers from psychology but also from social sciences like political sciences, sociology, and economics, and

then add references ONLY to economics papers??? Many aspects you are investigating have been investigated for even tiny subfields of the different disciplines of the social sciences and the sample sizes are often much bigger even for such studies. You just don't do a proper literature review and then try to convince your readers that you can contribute anything with just 200 articles that you chose with a randomization process that first made you analyze papers that totally fall out of the scope of your study?

It is my understanding that a reviewer's job is to help select between not so good and good contributions for publication and give advice on how to improve the latter. Not to help authors who are able to do this themselves to improve their manuscripts to make them acceptable for publication.

My judgment is already made from the comments above and I refuse to spend more time reading the manuscript again as I regard this inadequate use of a peer's voluntarily contributed time.

Given that also reviewer one already refused to look at the manuscript again under the circumstances I would advise the authors to prepare their research better before submission to academic journals. They are certainly able to do so as at least one of them has already published high quality inspiring research in the past. I am looking forward to reading more of that in the future. If you need any help or advice you know how to reach me. I am sorry I cannot be of more help at this point.

Author's Response to Decision Letter for (RSOS-190806.R1)

See Appendix B.

Decision letter (RSOS-190806.R2)

14-Jan-2020

Dear Dr Hardwicke,

It is a pleasure to accept your manuscript entitled "An empirical assessment of transparency and reproducibility-related research practices in the social sciences (2014-2017)" in its current form for publication in Royal Society Open Science.

Best regards,

Lianne Parkhouse
Editorial Coordinator
Royal Society Open Science

on behalf of Professor Chris Chambers (Associate Editor) and Professor Essi Viding (Subject Editor)

Follow Royal Society Publishing on Twitter: [@RSocPublishing](https://twitter.com/RSocPublishing)

Appendix A

Response to referees

Manuscript ID: RSOS-190806

Title: An empirical assessment of transparency and reproducibility-related research practices in the social sciences (2014-2017)

We are grateful to the editor and referees for their in-depth assessment of the manuscript and helpful feedback. Below we have responded to each comment in turn.

Editor/ reviewer	Comment number	Comment	Author response
Associate Editor 1	1	Three expert reviewers have now appraised this manuscript. The assessments are mixed overall. Reviewer 1 recommends rejection on the grounds that the article falls out of scope for RSOS. The issue of scope is an editorial judgment, and in sending the manuscript for in-depth review, the journal has already decided that the focus on reproducibility and open research practices places it within scope, similar to previous articles published in this journal. Therefore, as much as the journal is grateful for the assessment of Reviewer 1, the authors not need respond to this review. Reviewer 2 is also very negative for different reasons, chiefly that the inclusion criteria are unclear (or poorly chosen), and that a variety of methodological details are either unspecified or unjustified. Finally, Reviewer 3 is more positive but raises concerns with the selection of articles and sample size, echoing the concerns of Reviewer 2. Overall, the reviews are too negative to proceed with publication, and the manuscript falls very close to the line for outright rejection. However given the positive assessment of Reviewer 3, and the overall importance of this question, I recommend a comprehensive Major Revision. This is likely to require additional data collection and analysis, quite possibly in deviation from the preregistered protocol, and it may be that the authors will prefer to voluntarily withdraw their	We are grateful for the additional time granted to conduct further data collection and address the issues raised. We have revised the paper based on the results of the additional data collection and the reviewer's comments. Specific responses are detailed below. As instructed by the editor, we have not responded to the comments of Reviewer 1.

		submission rather than embark on the substantial extra work that is likely to be needed to satisfy Reviewers 2 and 3.	
Reviewer 2	2	The “Subject” and “Subject Category” of this study are reported as “: psychology < BIOLOGY” and Psychology and cognitive neuroscience” although as already the title reveals this study is about “transparency and reproducibility-related research practices in the _social sciences_”	This was simply the most applicable category label we could find in the journal submission system.
	3	Looking at the database used it surprises that it includes numerous studies from journals that do not or certainly not primarily cover social sciences. Methodologically focused journals like Canadian Journal of Statistics https://www.scimagojr.com/journals_earch.php?q=28893&tip=sid Computational Statistics https://www.scimagojr.com/journals_earch.php?q=28930&tip=sid Journal of Applied Probability https://www.scimagojr.com/journals_earch.php?q=23838&tip=sid Communications in Statistics Part B: Simulation and Computation https://www.scimagojr.com/journals_earch.php?q=23526&tip=sid Computational and Applied Mathematics https://www.scimagojr.com/journals_earch.php?q=5000153703&tip=sid Linear Algebra and Its Applications https://www.scimagojr.com/journals_earch.php?q=24475&tip=sid Mathematical Inequalities & Applications https://www.scimagojr.com/journals_earch.php?q=24572&tip=sid may still be acceptable. As the above links to the publicly available Scimago Journal Rank information of Scopus show, Scopus categorizes the first three of these journals also under “decision sciences”. The topics of the studies in several cases however are difficult to categorize as	We thank the reviewer for their close scrutiny of the data. The reviewer highlights a number of journals in which articles in the sample were published and says that those journals do not primarily cover social sciences research. That may be true, but it is not relevant because the articles were not defined as social sciences research by the journal they were published in, but by a citation clustering method (Klavens & Boyack, 2017). However, the reviewer’s comment led us to realize that (a) the sampling process was not described in sufficient detail in the manuscript; and (b) an error had occurred during the original sampling process. Below we provide a full description of the sampling process and describe how we have corrected the error. The manuscript contains a condensed version of this description and refers readers to an amended protocol (available here: https://osf.io/j5zsh/) for a full account of the steps taken to correct the error (see L81-108, p.6-8) The sample selection process is illustrated in Figure 1. The sample was drawn from a database that classifies academic articles according to one of 12 broad fields of science according to a model of the disciplinary structure of the scientific literature (Klavans & Boyack, 2017). The model was created using Scopus-indexed content from 1996 to April 2017 which was grouped into 91,726 clusters of documents using citation information. Each cluster was assigned to one of 12 broad fields of science. Details about this model and method are given in Klavans & Boyack (2017). Of the 91,726 clusters, 14,342 were assigned to the Social Sciences category. For the time period of interest (2014-2017), the 14,342 social science

	social sciences. In many other cases it is completely unclear how the studies could end up in a sample of articles from the social sciences. There are studies from journals like Plastics Engineering (the doi is missing, it is 10.1002/j.1941-9635.2015.tb01322.x and the Scopus link shows there is no connection whatsoever to the social sciences https://www.scimagojr.com/journals_earch.php?q=14353&tip=sid and even the database provided categorizes this as “Materials Chemistry”), CHEMICAL ENGINEERING TRANSACTIONS (Scopus and the provided database classify this journal under chemical engineering, not a subfield of social sciences: https://www.scimagojr.com/journals_earch.php?q=19600161818&tip=sid) , BMJ Open (https://www.scimagojr.com/journal_search.php?q=19800188003&tip=sid the study topic arguably falls under social sciences but the journal is classified as Medicine as one would assume from the abbreviation (British Medical Journal...)), Journal of Engineering for Gas Turbines and Power https://www.scimagojr.com/journals_earch.php?q=20962&tip=sid, Frontiers of Information Technology and Electronic Engineering https://www.scimagojr.com/journals_earch.php?q=21100409130&tip=sid European Journal of Paediatric Dentistry https://www.scimagojr.com/journals_earch.php?q=25027&tip=sid Carpathian Journal of Earth and Environmental Sciences	clusters contained 648,063 documents with an article type of ‘article’ or ‘review’ (as determined by Scopus). A random number generator was used to randomly order these documents, and the first 600 were selected. We subsequently decided to select 250 of these 600 articles for further analysis, based on our judgement of what was both informative and tractable. For example, for an expected proportion of 0.2 and a 95% confidence interval we would expect a margin-of-error of 0.05 for a sample size of 246¹. 215 of these articles met our eligibility criteria of being English language and 15 were used for piloting purposes, leaving a sample of 200 articles available for data extraction and coding. After completing the study, a peer reviewer noticed that some articles in the sample did not intuitively seem to be well characterized as originating from the “social sciences”. Upon subsequent investigation and communication with the researcher who had devised the science mapping/clustering method that we used (Kevin Boyack), we found that due to an unknown error, the original sample of 600 articles contained 192 articles that had not been classified as social sciences by the model (408 articles were correctly included). Correspondingly, of the sample of 200 articles included in data extraction, 65 had not been classified as social sciences by the model (135 articles were correctly included). Despite the error, we are confident that the 408 articles correctly classified as social sciences in the original sample of 600 are a random sample of the 648,063 social sciences documents available in the database as described above. To further ensure the face validity of articles included in the sample, we also additionally limited the sample to articles that had an All Science Journal Classification (ASJC) code related to the social sciences, specifically “Economics, Econometrics and Finance” (ECON), “Psychology” (PSYCH), “Business, Management and Accounting” (BUS), and “Social Sciences” (SOC). This reduced the sample of 408 articles to 332 articles. Correspondingly, the sample of 200
--	---	---

¹ This calculation was performed using the formula $N = P(1-P)(Z/E)^2$ where: N = sample size; Z = the value from standard normal distribution corresponding to desired confidence level (i.e., Z=1.96 for 95% CI); P = expected true proportion; and E = margin of error.

https://www.scimagojr.com/journals_earch.php?q=15900154727&tip=sid
Wounds UK

https://www.scimagojr.com/journals_earch.php?q=4500151403&tip=sid
Canadian Family Physician

https://www.scimagojr.com/journals_earch.php?q=110256&tip=sid
Indian Journal of Science and Technology

https://www.scimagojr.com/journals_earch.php?q=21100201522&tip=sid
Physical review. E

https://www.scimagojr.com/journals_earch.php?q=21100855841&tip=sid
(in this case surprisingly at least the subject of the study can be described as social science even though Scopus categorizes the journal under mathematics)

Journal of Uncertain Systems

https://www.scimagojr.com/journals_earch.php?q=19900191975&tip=sid
Journal of Intellectual Disability Research

https://www.scimagojr.com/journals_earch.php?q=16726&tip=sid
Fire Rescue Magazine

https://www.scimagojr.com/journals_earch.php?q=5000156910&tip=sid
Journal of Vocational Rehabilitation

https://www.scimagojr.com/journals_earch.php?q=29285&tip=sid
Journal of Clinical Urology

https://www.scimagojr.com/journals_earch.php?q=21100235629&tip=sid
European Journal of Philosophy

https://www.scimagojr.com/journals_earch.php?q=5600155103&tip=sid
Archives of Physical Medicine and Rehabilitation

https://www.scimagojr.com/journals_earch.php?q=16270&tip=sid
Journal of Nutrition and Health

https://www.scimagojr.com/journals_earch.php?q=21100259127&tip=sid
Zhongguo Jixie Gongcheng/China Mechanical Engineering

https://www.scimagojr.com/journals_earch.php?q=22181&tip=sid
Computers in Industry

https://www.scimagojr.com/journals_earch.php?q=19080&tip=sid

articles we previously used in the study contained 108 articles that met these criteria.

The 332 articles represent a random sample of the 485,460 documents from the database that were classified as social sciences and have one of the above ASJC codes. The number of documents broken down by ASJC code is as follows: BUS, 105,447; ECON, 92,348; PSYCH, 75,353; SOC, 324,618. After removing articles that were not English language (35) and those we had already coded (108), 189 new articles remained.

In order to meet our original sample target of 250 articles, we randomly sampled 142 of these new articles (using the R function *sample_n* from the package *randomizr*) and combined them with the 108 articles that had already been coded and met the above criteria. The final sample of 250 articles represents a random sample of the 485,460 English-language articles available in the database (Klavens & Boyack, 2017) that were classified into one of the social sciences clusters, also had an ASJC code specifically related to the social sciences (BUS, ECON, PSYCH, or SOC), and were published between January 2014 and April 2017.

	Journal of Musicology https://www.scimagojr.com/journals_earch.php?q=14000155925&tip=sid Pacific Historical Review https://www.scimagojr.com/journals_earch.php?q=23676&tip=sid (study topic arguably social science but journal categorized as “arts and humanities – history”) Statistics in Medicine https://www.scimagojr.com/journals_earch.php?q=20086&tip=sid twice The Chaucer Review (correctly categorized as “literature” in the database) https://www.scimagojr.com/journals_earch.php?q=13243&tip=sid International Journal of Developmental Neuroscience https://www.scimagojr.com/journals_earch.php?q=16147&tip=sid Lecture Notes in Computer Science https://www.scimagojr.com/journals_earch.php?q=25674&tip=sid twice Revista Facultad de Ingenieria https://www.scimagojr.com/journals_earch.php?q=12400154740&tip=sid (in one case a social science topic but the journal is not categorized as social science) Biogeosciences https://www.scimagojr.com/journals_earch.php?q=130037&tip=sid Currents in Pharmacy Teaching and Learning https://www.scimagojr.com/journals_earch.php?q=19500157042&tip=sid Diabetes Primary Care https://www.scimagojr.com/journals_earch.php?q=5200152617&tip=sid SMT Surface Mount Technology Magazine https://www.scimagojr.com/journals_earch.php?q=27175&tip=sid If these studies were indeed found using a Scopus database this may be useful to illustrate that this Scopus database is not very useful in identifying social science research. The data presented is certainly not useful to make assessments about the state of transparency in the	Figure 1. Flow diagram illustrating the sample selection process. Full details are provided in the main text.
--	---	--

		social sciences. To be honest I am not even sure if this may just be a test whether reviewers actually look at underlying data of a study if they have the chance because based on the data that is made available this analysis does not make any sense at all. Unfortunately, I had already used some of my time to start a referee report under the assumption that this was a serious analysis. I leave it in the current state as given what I saw in the database I see no value in writing a full report and strongly recommend a rejection of this study.	
	4	In the article it is claimed that the study was pre-registered. In the pre-registration report, it is stated: "Of the 215 eligible articles, we have randomly selected 15 to be used for piloting purposes, leaving 200 eligible articles in the main sample". This statement shows that the pre-registration already included the result of the sampling process.	Thank you for this comment, however we believe the pre-registered status of the study protocol has been accurately communicated. The article states that the study protocol was pre-registered: The study protocol was pre-registered on July 3, 2018 and the study protocol states that the sample had already been obtained: From Scopus we have randomly sampled 250 articles published between January 2014 and April 2017. It is not clear to us if the reviewer is suggesting this is problematic (we do not see how it is). Note that considering that our sampling process was amended as outlined in our response to comment #3, comment #4 may no longer be applicable.
	5	It is furthermore not explained how the randomization was conducted	Thank you for drawing our attention to this omission. As noted in response to comment #3, we have provided more detail about the sampling process including the following which explains how the random sampling was performed: Random sampling was performed by using a random number generator to shuffle the order of the articles and selecting the top N articles required.
	6	and the piloting purposes are not motivated or explained in any way.	The pilot was designed to see if the data extraction procedures worked in practice. To the procedure section of the article we have added: We pilot tested the data extraction procedures using 15 articles that were not included in the final sample.
	7	It is not defined what "raw data" exactly means in the study. Given the big differences in study designs it would have been necessary to define clearly in advance what kind of raw data is expected in which case (and why). The policy of a number of	All of the measured variables presented in Table 1 have specific operational definitions which were used by the coders during data extraction. Table 1 refers readers to the data extraction form for this information: Further details about extraction and coding is available here: https://osf.io/v4f59/ We have now amended this for clarity: The exact

		journals that raw data need to be made available to other researchers on request is not mentioned.	operational definitions and procedures for data extraction/coding are available in the structured form here: https://osf.io/v4f59/ To answer the specific point about raw data, in the form section on data it is stated that: "Data" refers to recorded information that supports the analyses reported in the article. For our purposes, we use "data" synonymously with "raw data" meaning recorded information in its rawest, digital form, at the level of sampling units (e.g., participants, homes, companies, etc). A "data availability statement" can be as simple as a url link to a data file, or as complex as a written explanation as to why data cannot be shared.
	8	It is not explained how it was checked if a study had been replicated, been part of a systematic review or a meta-analysis. Which citation databases were investigated? Scopus again?	As noted in our response to comment 7, all detailed operational definitions are available in the extraction form. As stated there, we used the Scopus database to identify citing articles. We then screened articles at the title level, abstract level, or full text level as necessary in order to determine their status with regard to replication/systematic review/meta-analysis.
	9	What exactly is meant with "2016 journal impact factor"? 2-year-SSCI or Scimago Journal Rank? The latter would have been more appropriate to report given that the studies were selected from Scopus, which includes much more journals than the SSCI. It seems more likely that SSCI was used given that it is reported many of the analyzed studies were published in journals for which no impact factor was available.	As noted in our response to comment 7, all detailed operational definitions are available in the extraction form. As stated there, we used Thomson Reuters Journal Citation Reports to identify the 2016 journal impact factor. We have also now clarified this in Table 1 of the manuscript as follows: 2016 journal impact factor (according to Thomson Reuters Journal Citation Reports)
	10	To assess how big the sample drawn for the study is compared to the whole distribution of social science studies in the SCOPUS database it should be stated how many studies were classified as social science in Scopus in 2014-2017 altogether. Without this number the reported "95% confidence intervals based on the Sison-Glaz method for multinomial proportions" (that I am not familiar with) are not convincing.	Thank you for drawing our attention to this omission. As noted in our response to comment 3, we now provide a more extensive description of the sampling procedures. Specific to this point, we now state (L105-108, p.7-8) that The final sample of 250 articles represents a random sample of the 485,460 English-language articles available in the database [27] that were classified into one of the social sciences clusters, also had an ASJC code specifically related to the social sciences (BUS,

			ECON, PSYCH, or SOC), and were published between January 2014 and April 2017.
11	Altogether it seems questionable how such a small sample can help to draw conclusions on research in the social sciences in general.	The target sample size of 250 articles was based on our judgement of what sample size would be sufficiently large enough to be informative but also realistically feasible for manual data extraction. For example, for an expected proportion of 0.2 and a 95% confidence interval we would expect a margin-of-error of 0.05 for a sample size of 246. [This calculation was performed using the formula $N = P(1-P)(Z/E)^2$ where: N = sample size; Z = the value from standard normal distribution corresponding to desired confidence level (i.e., Z=1.96 for 95% CI); P = expected true proportion; and E = margin of error.] The reviewer's comment has prompted us to add to the discussion section a limitation of the study that we had not previously made explicit (L275-279, P.17-18): Fourthly, although our sample can be used to estimate the prevalence of the measured indicators broadly in the social sciences, those estimates may not necessarily generalize to specific contexts, for example specific sub-fields or articles published in specific journals. It is known for example, that specific journal policies can be associated with increases in data and materials availability [6,25,26].	
12	It is not investigated how transparency measures vary across journals of different rank, and with such a small sample size this should be impossible. Why did the authors then not use available data from many previous studies that analyzed the many aspects mentioned instead?	Although we collected data on journal impact factors for descriptive purposes, it was not our intention to draw inferences about their association with the indicators we measured.	
13	One additional detail that was neglected in the analysis: Focusing on journals categorized as "social science" in Scopus excludes all studies from the social sciences that were published in journals like Nature, Science, the Proceedings of the National Academy of Sciences or Royal Society Open Science that cover all sciences. At least this should be mentioned and the importance of such studies in	We acknowledge in the revised description of the sampling procedures (L94-97, p.7) ...this means the sample would not capture social science articles published in multidisciplinary journals (e.g. Nature, Science, PNAS, RSOS) and/or journals that belong mainly to other disciplines	

		comparison to all social science studies should be assessed. While the number will be relatively slow, the impact of such studies will on average be much higher than that of most others because they were deemed important enough to be published in such general interest journals.	
	14	The variable “country of origin” is not defined: Of the data? Of the authors or their affiliations? If the latter, how were cases with more than one author treated?	As noted in our response to comment 7, all detailed operational definitions are available in the extraction form. We have clarified in Table 1 of the manuscript that this variable is referring to the location of the corresponding author’s affiliation: country of origin (based on corresponding author’s affiliation)
	15	The “wide range of journal impact factors (median 1.33, range 0.25-16.79; based on 2016 journal impact factor” should be compared to the whole distribution of thousands of journals. Studies from how many journals out of a distribution of how many total social science Scopus journals were investigated? How does the distribution of journal impact factors in the sample compare to the distribution of impact factors of all Scopus social science journals? For just some thousand journals these informative descriptive statistics are not difficult to obtain. The same holds for values of the whole distribution for subject areas and years of publication.	Thank you for this suggestion. However, we do not think such a comparison would be particularly informative – articles in our sample were characterized as ‘social sciences’ based on a citation clustering method (see response to comment 3). Thus, it is not just an issue of how many journals would be categorized as publishing in the social sciences, but also how many articles they published which are indeed in the social sciences, and we would have to calculate new JIFs for thousands of journals on this basis. We believe this is beyond the scope of our current evaluation. In the new description of the sampling procedures we state that (L96-98, p.7) ...the number of documents in the database broken by ASJC code was as follows: BUS, 105,447; ECON, 92,348; PSYCH, 75,353; SOC, 324,618
	16	The judgment “articles associated with empirical data were rarely cited at all (median = 1, min = 0, max = 31)” does not seem justified given the reported median journal impact factor of 1.33 – how is 1 low as compared to 1.33?	We are afraid that the meaning of the comment is not completely clear to us. The statement, articles associated with empirical data were rarely cited at all , is an absolute one, not a relative one. In other words, we are not claiming the number of citations is rare relative to the journal impact factor, but that the number of citations is rare.
	17	From which databases were citing articles identified and when?	The database was Scopus (please see response to comment 8). We now note in the procedure section that (L118-119, p.8): Data collection took place between May 7th, 2018, and October 10th, 2019.
	18	Given that the sample includes studies published in 2017 and it seems likely that the cited journal	We have incorporated this sentiment into a sentence in the discussion which already acknowledges the limitation of the citation history

		impact factors refer to the last two years, it is not surprising that the empirical studies had not yet been cited so often by at the latest May 2019.	assessments (L256-259, p. 17): We assessed article citation histories in order to gauge how often they had been cited overall and cited by replication studies, meta-analyses, or systematic reviews specifically. It should be noted that our sample pertained to recently published studies and it may take some time before studies that build upon the original articles are themselves published.
19		In the conclusion it is pointed out that “social science research often addresses topics that are highly pertinent to policy makers and policy decisions based on flawed research can have substantial economic, social, and individual costs.” This is illustrated as follows: “a paper entitled “Growth in a Time of Debt” which influenced government austerity policies around the world was found to contain a serious analytic error by a student examining formulas and data in the original spreadsheet.” While both statements are true, to analyze the impact of the lack of transparency on policies it is necessary to investigate papers that really have an impact and not a random sample of studies covered in the Scopus database. As is well known, most academic studies are hardly ever cited even in academia and certainly do not have any impact beyond. The study “Growth in a Time of Debt” was published in 2010 in a journal with one of the highest impact factors in the social sciences and had already been cited more than a thousand times by 2014. To argue about the impact of lack of transparency in social science one needs to analyze the transparency of impactful social science studies, which are typically published in more cited journals, which tend to have much stronger transparency rules than the average journal. In addition, to assess the impact on policy one may not focus on citations in academia but in policy papers or in the non-scientific media.	Thank you for this comment. It was not the purpose of the study to investigate “the impact of the lack of transparency on policies” but to investigate the prevalence of indicators of transparency and reproducibility. In the discussion, we emphasize several reasons why these are important principles, one of which is the potential impact on downstream policy making. We say that (L202, P.14): Poor transparency *can* have very real costs (emphasis added) and do not claim that the low transparency in the current sample *will* have these downstream consequences - that is unknown. The Growth in the Time of Debt study is used as an illustration of how (L205-206, P.17): transparency and sharing can have tangible benefits.

20	Similarly, when discussing the state of open access it should be taken into account that in social sciences like economics none of the most highly ranked journals are published open access or even give the option to publish papers open access, so open access has so far been a phenomenon mostly for low prestige research. It would be interesting to see how this varies across sub-disciplines, time and journal ranks, but with a sample of 200 randomly selected studies over three years it will be impossible to make a real assessment of any of these aspects.	Our findings provide an estimate of the prevalence of transparency and reproducibility indicators in the social sciences. The extent to which the findings generalize to any specific context is unknown and needs careful consideration of relevant factors influencing that context. Unfortunately, we cannot address all possible interesting contexts in a single study. As noted in the response to comment 11, we have added to the discussion a limitation of the study that addresses this point directly (L275-279, P.17-18): Fourthly, although our sample can be used to estimate the prevalence of the measured indicators broadly in the social sciences, those estimates may not necessarily generalize to specific contexts, for example specific sub-fields or articles published in specific journals. It is known for example, that specific journal policies can be associated with increases in data and materials availability [6,25,26].
21	In the discussion of the advantages of preregistration an important point is left out: In social sciences where data is often readily available there is no way to know whether a “pre-registration” was actually registered before or after an analysis (with potential p-hacking and HARKing) was done.	The point that when data is readily available there is “no way to know whether a ‘pre-registration’ was actually registered before or after an analysis” also applies to studies that involve primary data collection, so we do not think that it is especially pertinent to discuss this point in the present article.
22	The discussion of possible reasons for lack of transparency includes the following statement: “It is possible that a lack of transparency is well justified for some of the articles we have examined. However, such justifications were not made explicit.” Numerous social science journals publish reports about data availability, including cases of sensitive or proprietary data, and there are guidelines how to inform about such restrictions. If none such case was in the sample this just reveals that the chosen sample size was too small to allow an informative assessment of the state of transparency in the social sciences.	We are afraid we are not sure what is meant by “Numerous social science journals publish reports about data availability”. Our assessment was of data availability statements provided in the articles in the sample. It seems a reasonable expectation to us that an article making a scientific claim should either provide the data that underlies that claim or state in the article why the data cannot be shared (also see Morey et al. 2016 https://doi.org/10.1098/rsos.150547) We have addressed the points about the informativeness of the sample in our response to comment 11.
23	Charitè is spelled with a wrong accent (should be Charité).	Thank you for catching this error.

	24	As a last comment, the authors heavily cite previous studies from just one social science, psychology, many of whose sub-disciplines share more characteristics with life sciences than with other social sciences. The one exception is a study from economics that is used as an example, which does not suit its purpose well. It would be beneficial to the study to draw on the existing literature on transparency in the social sciences from fields like political sciences, sociology, and economics.	We agree with the reviewer's suggestion and have added the following references: Wood, B., Müller, R., Brown, A. (2018). Push button replication: Is impact evaluation evidence for international development verifiable? PLOS ONE 13(12), e0209416. https://dx.doi.org/10.1371/journal.pone.0209416 Franco, A., Malhotra, N., Simonovits, G. (2014). Publication bias in the social sciences: Unlocking the file drawer Science 345(6203), 1502-1505. https://dx.doi.org/10.1126/science.1255484 McCullough, B., McGeary, K., Harrison, T. (2008). Do economics journal archives promote replicable research? Canadian Journal of Economics 4(), 1406-1420. Chang, A., Li, P. (2015). Is Economics Research Replicable? Sixty Published Papers from Thirteen Journals Say "Usually Not" Finance and Economics Discussion Series 2015(83), 1-26. https://dx.doi.org/10.17016/feds.2015.083 Olken, B. (2015). Promises and Perils of Pre-Analysis Plans Journal of Economic Perspectives 29(3), 61-80. https://dx.doi.org/10.1257/jep.29.3.61 Krawczyk, M., Reuben, E. (2012). (Un)Available upon Request: Field Experiment on Researchers' Willingness to Share Supplementary Materials Accountability in Research 19(3), 175-186. https://dx.doi.org/10.1080/08989621.2012.678688 Camerer, C., Dreber, A., Forsell, E., Ho, T., Huber, J., Johannesson, M., Kirchler, M., Almenberg, J., Altmejd, A., Chan, T., Heikensten, E., Holzmeister, F., Imai, T., Isaksson, S., Nave, G., Pfeiffer, T., Razon, M., Wu, H. (2016). Evaluating replicability of laboratory experiments in economics Science 351(6280), 1433-1436. https://dx.doi.org/10.1126/science.aaf0918 Necker, S. (2014). Scientific misbehavior in economics Research Policy 43(10), 1747-1759. https://dx.doi.org/10.1016/j.respol.2014.05.002
--	----	---	--

Reviewer 3	25	There is a lot to like about this manuscript, and I am broadly supportive of publication, provided that the authors are able make some minor revisions. The manuscript provides an important snapshot of the state of transparency practices in the social sciences today. Figure 1 is destined to be widely commented on, and taught in courses. The fact that the analysis is pre-registered is a major strength. Overall, I find the analysis credible and the findings innovative. I would suggest a few possible edits to the authors.	We appreciate reviewer #3's support and helpful comments.
	26	(1) The authors should comment more on the small sample size of articles included in the analysis (N=250 initially, which falls to 198 in the final analysis due mainly to an English language exclusion criterion). Given the important of this topic, and the relatively low cost of classifying articles, why such a small sample? A larger sample would produce a more definitive statistical analysis. This is an important limitation of the current analysis that should be flagged for the reader.	Thank you for this suggestion. We believe that the sample size provides a reasonable balance of informativeness and tractability (please see response to reviewer 2 comment 11 for more details). We note that, after correcting the error noted in the response to comment 3, we ensured that only English language articles were sampled, thus increasing the eligible sample size from the previous version by 52 articles. Estimates are accompanied by confidence intervals which readers can use to gauge precision. We think the level of precision is very reasonable - for example, taking the first sentence reporting results in the abstract (L31-33, P.2): Few articles indicated availability of materials (16/151, 11% [95% confidence interval, 7% to 16%]), protocols (0/156, 0% [0% to 1%]), raw data (11/156, 7% [2% to 13%]), or analysis scripts (11/156, 7% [2% to 13%]), and no studies were pre-registered (0/156, 0% [0% to 1%]). The classification process, which was performed manually and in duplicate, required a substantial amount of time so increasing the sample size further would come with significant costs. It was necessary to recruit two additional team members to assist with the further data collection involved in this revision process.

	27	(2) The authors (rightfully) point out that, given their recent publication dates, few of the articles could reasonably already be included in meta-analyses or systematic reviews. Given this, I would remove presentation of these results from the main figure and text, and instead relegate them to the appendix. This would save space in the main article.	As the reviewer notes, we are in agreement about this limitation of the citation history data and explicitly acknowledge this in the manuscript. However, we do not think these data are sufficiently uninformative such that they be relegated to an appendix. Note that it is already the case that citation history outcomes are not presented in the figure.
	28	(3) There is too little detail on the nature of the social science articles that are considered in the analysis. If the authors seek to inform a social science research community audience, they will need to do a better job of characterizing the sample of journals, scholars, articles, and topics that they focus on here. One interpretation is that the paper was written by medical researchers with limited knowledge of the “terrain” they are operating in here, in the social sciences. The authors inform us that very few of the articles, only 17 out of 198, are in Economics, Political Science, or Sociology, three of the largest, best cited and most influential social science fields for public policy. This is a disappointment, since it feels (to me) like this sample of articles may not be very representative of influential social science research. Economics and Political Science are also social science fields with some of the most visible open science “movements”. Some specific questions. What fields are the N=125 articles that are in “other sub-areas” actually in? The fact that around 40% of articles in the sample also have “no empirical data” leaves me wondering what types of studies these are. There are many possibilities – theoretical / conceptual work, ethnographies, historical work (without original data), critical theory – but which of	We appreciate the reviewer’s comment and the opportunity to respond to these concerns. (1) Firstly, we note that there have been some changes to the sample since the previous version of the manuscript due to the discovery of an error – the steps taken to address the error are detailed in our response to comment 3. (2) Regarding the concern about representativeness, we would like to re-emphasize that the goal of the study was to (L66-68, P.4): estimate the prevalence of a range of transparency and reproducibility-related indicators in the social sciences literature published between 2014-2017 – a sentence we have modified in the introduction for clarity and now re-iterate in the abstract. Thus, it is not necessarily the case that our findings generalize straightforwardly to any specific context in which particular factors may influence those estimates. For example, the reviewer highlights that the findings may not be representative of “three of the largest, best cited and most influential science fields for public policy”. Indeed, this is unknown, remains an empirical question, and was beyond the scope of the present study. We are well aware that some sub-fields may be performing better on the measured indicators and are in fact currently conducting a study similar to the present one in the domain of psychology, in which many ‘open science’ initiatives have been introduced. Even within the broad domain of psychology though, there will be contexts where transparency will be more or less prevalent (experimental psychology vs clinical psychology, articles published in journals with data sharing policies vs those without data sharing policies). Unfortunately, all of these interesting contexts cannot be addressed in a single study. We hope

	these are in the sample being considered here? Overall, this came across to me as the most important limitation of the article. I do think that having a larger sample size of articles, and perhaps stratifying the random sampling of articles in a more systematic way – perhaps oversampling articles in journals with higher impact factors, or making sure you have at least some minimal representation of major social science fields, i.e., more than just 8 articles in Sociology and Political Science combined! – would go a long way in future work along these lines.	that our addition of the following limitation to the discussion section makes the scope of the study clearer (L275-279, P.17-18): Fourthly, although our sample can be used to estimate the prevalence of the measured indicators broadly in the social sciences, those estimates may not necessarily generalize to specific contexts, for example specific sub-fields or articles published in specific journals. It is known for example, that specific journal policies can be associated with increases in data and materials availability [6,25,26]. (3) In response to the reviewer’s specific question “What fields are the N=125 articles that are in “other sub-areas” actually in?” There is a table footnote which states that (L132, P.10) For all subject areas see https://osf.io/7fm96/ We do not think displaying all additional 44 rows of this part of the table in the main text would be an effective use of space and the information is readily available in the linked file. (4) Regarding articles classified as “no empirical data”, Table 1 notes that specific operational definitions used by the coders are available in the data extraction form (https://osf.io/v4f59/). In this case, examples of “no empirical data” would be editorials, commentaries [without reanalysis], simulations, news, and reviews. We did not record more detailed information about study designs with this classification type.
--	--	--

Appendix B

Response to referees

Manuscript ID: RSOS-190806

Title: An empirical assessment of transparency and reproducibility-related research practices in the social sciences (2014-2017)

We are grateful to the editor and referee for their continued in-depth assessment of the manuscript and helpful feedback. We have uploaded a revised version of the manuscript (“manuscript_postReview_clean”) which should be considered the primary version. For convenience we have uploaded the same revised version with tracked changes from the previously submitted version (“manuscript_postReview_trackChanges”). Below we have responded to each comment in turn.

Editor/ reviewer	Comment number	Comment	Author response
Reviewer 2	1	Subject:psychology < BIOLOGY and Subject Category:Psychology and cognitive neuroscience still do not fit. If the journal chooses to accept social science contributions, appropriate categories should be offered to submitting authors.	As mentioned in our previous response, the category options are outside of our control and we chose the closest fit available to us.
Reviewer 2	2	I find a bit confusing that there are two different files for the text, one apparently with changes from the original manuscript in red color and the other one the revised manuscript but I could not find this explained anywhere.	We are sorry this was not clear. We uploaded a revised version of the manuscript (“manuscript_postReview_clean”) and the same revised version with “track changes” on (“manuscript_postReview_trackChanges”) to help the editor/reviewers identify all of the changes we had made. We now explain this at the top of this “response to referees” document.
Reviewer 2	3	In my first round review I commented that a large part of the studies analyzed were taken from journals that do not primarily cover social sciences, that were classified as covering different fields by the Scopus database that the authors used and that a number of the studies had topics that did not resemble social science research. The authors reduce this comment to “The reviewer highlights a number of journals in which articles in the sample were published and says that those journals do not primarily cover social sciences research.” They claim this is “not relevant [seriously] because the articles were not defined as social sciences research by the journal they were published in, but by a	We remain grateful to the reviewer for the careful observation that led us to discover an error in the original version that we then fixed in the previous revision. The core issue still being raised is how one should operationalize the process of obtaining a sample of articles from, in this case, the social sciences. In response to the reviewer’s previous comments, we made adjustments to the sampling procedures and we added a more detailed statement of the sampling procedures to the manuscript. To briefly reiterate the key points, the boundaries between different disciplines are overlapping and fuzzy, not definitive. There is no single ‘correct’ way to obtain a sample of articles from a particular field – there are difference ways to do this, each with limitations. We initially relied upon the citation clustering method employed by Klavans & Boyack (2017) to identify social sciences articles. This has the advantage of operating at the article

		citation clustering method". I cannot help but ask the authors: If you need a reviewer to see that these engineering, literature, philosophy, medicine or chemistry studies have nothing whatsoever to do with social sciences, how come you feel qualified to do research about the social sciences?	level and is therefore sensitive to social sciences articles that may not have been published in journals classified as "social sciences" by databases like Scopus (for example, an article published in a multidisciplinary journal, like Science). However, one disadvantage of this method is that it might also capture some articles that are cited in the social sciences literature, but are not themselves social sciences research – for example an article about a widely used statistical method. Thus, as noted in our previous response: "To further ensure the face validity of articles included in the sample, we also additionally limited the sample to articles that had an All Science Journal Classification (ASJC) code related to the social sciences...". In other words, the articles in the sample have been classified as belonging to the social sciences by two independent methods. Note that, as stated in the manuscript "this means the sample would not capture social science articles published in multidisciplinary journals (e.g., Nature, Science, PNAS, RSOS) and/or journals that belong mainly to other disciplines."
Reviewer 2		I find embarrassing to read you even contacted another researcher to confirm that you had included many of the studies in your sample based on a sample of his in which he apparently had not even classified these studies as social science as you claimed. How come you could not figure this out yourself? How come that the origin of such an error can remain "unknown" in a pre-registered, supposedly well documented study on transparency and reproducibility?	The cause of the error in the initial sampling procedure remains unknown, however we are confident it has been addressed (see below). We initially requested a sample of 600 articles classified as 'social sciences' from Boyack, who selected them from the Klavans & Boyack (2017) database and sent them to us. The original data file did not have a column containing the article classifications – this seemed unnecessary because all of the articles should have been classified as social sciences. Only after we contacted Boyack following the last round of peer review did he discover that this original sample contained some articles that had not in fact been classified as social sciences. We take full responsibility for the error, we will introduce additional checking procedures in future projects to make sure it does not happen again, and we are grateful to the reviewer for the careful assessment which led to its discovery. We are confident we have corrected the error because Boyack has provided classification information for all of the articles in the sample, so we know which were classified as social sciences and which were not. As noted above, we have also applied an ASJC filter to further ensure the validity of the sample.

Reviewer 2		It would certainly also be interesting to analyze what kind of social science research is published in journals that do not primarily have a social science focus. This aspect should however have been made clear to the reader from the start. Additional aspects should then be considered as for example how do they differ from standard social science contributions – there will be reasons why authors address a different audience. Given that the authors themselves choose for other parts of their study standard approaches looking at Scopus classifications by type of research or Thomson Reuters impact factors I find the mixed sampling method surprising and it is not clear to me why this approach was chosen.	This suggestion is no longer relevant as the ASJC filtering mentioned above means that all articles in the sample are from journals classified as “social sciences”. Of course, not everyone would agree 100% on any proposed classification. However, using a standard classification like ASJC minimizes subjectivity in judging what should qualify as social science.
Reviewer 2		I cannot follow the authors’ calculation for the sample size they regard as sufficient and cannot help but find the size of 200 studies for all social sciences together and for the number of aspects investigated alarmingly small.	If one wants to estimate the prevalence of some indices in a large population it is common practice to evaluate those indices in a sample and then use inferential methods (in this case confidence intervals) to gauge the precision of the obtained values as estimates of population parameters. When planning a study, one must consider resource constraints (in our case mainly personnel time) as well as informational value. A target sample size of 250 articles is a reasonable balance of information gain and tractability. In our previous response, we illustrated why 250 articles was a likely to be an informative sample size by performing a formal precision analysis (e.g., Rothman & Greenland, 2018; https://doi.org/10.1097/EDE.0000000000000876). As we noted in our previous response, now that the study is complete, what is relevant is the actual precision of the estimates. This can be gauged directly by examining the confidence intervals provided throughout the text. The obtained precision seems adequate to us; for example from the abstract: Few articles indicated availability of materials (16/151, 11% [95% confidence interval, 7% to 16%]), protocols (0/156, 0% [0% to 1%]), raw data (11/156, 7% [2% to 13%]), or analysis scripts

			(11/156, 7% [2% to 13%]), and no studies were pre-registered (0/156, 0% [0% to 1%]).
Reviewer 2		The data provided under the link https://osf.io/3d5um/ includes only 250 studies. It is not clear at first sight which ones were among the 108 from the previous version of the study, why in a study on transparency and reproducibility the 65 that had originally been miscategorized and the 27 that were later disregarded because of the added journal classification criterion as well as those that were disregarded as not in the English language are hidden now, and which ones were the 15 “used for piloting purposes”. Why is the full sample of 600 studies not shown, especially by researchers who write they use “raw data” as synonymous to “data”?	We have now created a separate “Error Documentation” document which we have uploaded to the OSF (https://osf.io/7anx6/) and referenced in the manuscript. This document contains an updated flow diagram illustrating the sample selection process and error correction process. At every stage where data are filtered, we provide a link to corresponding data files on the OSF.
Reviewer 2		Looking at the new data I find it questionable to consider studies on chemistry and nursing (education) or literary history in an analysis of transparency in the social sciences. But studies that are already categorized in your own data as philosophy and arts and humanities?	As noted above, the boundaries of scientific disciplines are fuzzy and overlapping. All of the articles included in the sample have been designated as social sciences though two independent operationalizations (Klavens & Boyack citation clustering method and ASJC) that were already available before we started our study, thus avoiding subjective choices on our part. By contrast, the reviewer is relying solely on a subjective assessment which has unknown operational characteristics and may miss important features of the articles which led them to be characterized as social sciences by the above methods. For example, the reviewer refers to “studies on chemistry”. We can only find one article related to chemistry in our sample – ID code “Xwpuk”. Based on the title alone - “Separation of caffeine from beverages and analysis using thin-layer chromatography and gas chromatography - Mass spectrometry” – one may indeed think this has nothing to do with the social sciences. However, the article is published in the Journal of Chemical Education and pertains to a study of class exercises in the context of chemistry experiments. The study is thus within the domains of chemistry and education , and one can see why although it may not be a prototypical example of social sciences research, it is reasonably classified

			as social sciences by both the citation clustering and ASJC methods.
Reviewer 2		I see that one has to limit one's research in some ways, just as an aside I'd like to note that the Scopus citation database that was used here is very incomplete even compared to readily available other data as for example that of google scholar.	Every database has its own coverage and of course not all journals are captured by every database. Scopus has the advantage of being widely used and being explicit about what it covers; social sciences are typically well covered in Scopus. Conversely, Google Scholar may have greater coverage in some domains but does not publicly disclose which journals are covered and how they are selected for inclusion. We are also aware of evidence which suggests that Google Scholar may be problematic when used for systematic searches due to inconsistent search returns (https://doi.org/10.1096/fj.07-9492LSF). Nevertheless, we have added another limitation in in the Discussion section: "Firstly, no database has perfect coverage of all journals in every field, but Scopus coverage is extensive. Google Scholar may have broader coverage than Scopus, but it is less transparent than Scopus about what journals and sources of information are included."
Reviewer 2		The choice to use the corresponding author's location as "country of origin" of a study does not convince me. Many studies nowadays are written by multinational teams and the corresponding author's location is not necessarily informative about where the most important parts of the research were performed.	This information is provided to illustrate the demographic characteristics of the sample. We do not draw any inferences based on the country of origin variable and do not suggest that the corresponding author's location indicates "where the most important parts of the research were performed." Note that in the manuscript we say - country of origin (based on corresponding author's affiliation) – so we believe it is quite clear what this variable refers to.
Reviewer 2		It is unclear to me how the word "rarely" could ever be meant in an "absolute" rather than a relative sense. I would say I have rarely read a study on social sciences that investigated philosophy and theology studies. In geological terms once in a human lifetime would usually not be considered a rare event. Analogously, one citation per study is not "rare" in bibliometric terms where the median and mode citation count is 1 or even 0 in most social sciences unless the sample of studies is restricted in particular ways.	We have changed "rarely" to "infrequently".
Reviewer 2		I continue to see the example of the Growth in a Time of Debt article	We do not suggest that "Growth in a Time of Debt" is representative of our sample but merely use it to

		as grossly misleading about the importance of the present study as it is not at all representative for a sample of randomly selected studies from all social (and some not so social) sciences. This should be made clear and not just be left to the reader's interpretation.	briefly illustrate the potential benefits of transparency (it is one of few well documented cases).
Reviewer 2		I maintain the point that when discussing the benefits of pre-registration also the shortcoming should be noted that when data is readily available as is most often the case in the social sciences it is impossible to know whether an analysis was actually done before or after a "pre"-registration. This does not apply in cases where data collection requires large funds that can usually only be acquired with a previously registered plan as is often the case in randomized controlled trials in, for example, medicine or development economics. Here pre-registration gives studies much more credibility but this cannot be generalized to the social sciences.	If scientists are stating in pre-registered protocols that they have not already obtained and/or analyzed data when in fact they have, then this would be outright fraudulent. It is not reasonable to expect pre-registration to prevent such outright fraud in any research domain. We think this discussion is important and similar debates have arisen in other observational fields (e.g. epidemiology), but it is probably tangential to the findings of the present study. Nevertheless, we have added to the discussion: "Additionally, pre-registration may be less pertinent when analyses of pre-existing data are intended to be entirely exploratory and no pre-conceived protocol really exists."
Reviewer 2		I also maintain my point that if the authors did not find a single case in which data was not made available because for legitimate reasons sensitive or proprietary data was used and this is explained by the journal this illustrates that the chosen sample was too small to allow an informative assessment of the state of transparency in the social sciences. At the very least informative parts of the literature should be cited and summarized that inform about the many journals that have specific rules about such data that cannot easily be shared.	This point seems to be entirely speculative and the empirical data we have obtained suggests that such cases are either not as frequent as the reviewer proposes or such justifications are not being stated in articles. We already refer to this in the discussion: "Thirdly, achieving transparency is not always straightforward when there are overriding legal, ethical, or practical concerns [43]. It is possible that a lack of transparency (in particular, a lack of data sharing) is well justified in some cases. However, no such justifications were stated in the articles we examined."
Reviewer 2		How can you agree that your literature review should not exclusively contain papers from psychology but also from social sciences like political sciences, sociology, and economics, and then add references ONLY to economics	Regarding existing research in sub-domains and the suggestion that we are not contributing anything - we reiterate that the goal of the present study was to estimate the prevalence of a range of indicators related to transparency and reproducibility broadly in the social sciences. We explicitly point out in the

		papers??? Many aspects you are investigating have been investigated for even tiny subfields of the different disciplines of the social sciences and the sample sizes are often much bigger even for such studies. You just don't do a proper literature review and then try to convince your readers that you can contribute anything with just 200 articles that you chose with a randomization process that first made you analyze papers that totally fall out of the scope of your study?	manuscript that the obtained estimates may not generalize to specific sub-domains. Regarding references – we have added some additional references from the mentioned domains (below), but we note that it is not our intention to, nor necessary to, provide an exhaustive literature review in the present context. Camerer, C.F., Dreber, A., Holzmeister, F. et al. Evaluating the replicability of social science experiments in Nature and Science between 2010 and 2015. Nat Hum Behav 2, 637–644 (2018) doi:10.1038/s41562-018-0399-z Stockemer, D., Koehler, S., & Lentz, T. (2018). Data Access, Transparency, and Replication: New Insights from the Political Behavior Literature. PS: Political Science & Politics, 51(4), 799-803. doi:10.1017/S1049096518000926 Zenk-Möltgen, Wolfgang & Lepthien, Greta. (2014). Data sharing in sociology journals. Online Information Review. 38. 709-722. doi:10.1108/OIR-05-2014-0119. Eubank, N. (2016). Lessons from a Decade of Replications at the Quarterly Journal of Political Science. PS: Political Science & Politics, 49(2), 273-276. doi:10.1017/S1049096516000196
Reviewer 2		It is my understanding that a reviewer's job is to help select between not so good and good contributions for publication and give advice on how to improve the latter. Not to help authors who are able to do this themselves to improve their manuscripts to make them acceptable for publication. My judgment is already made from the comments above and I refuse to spend more time reading the manuscript again as I regard this inadequate use of a peer's voluntarily contributed time. Given that also reviewer one already refused to look at the manuscript again under the circumstances I would advise the authors to prepare their research better	We thank the reviewer for their time and feedback. We are grateful for the detection of an error in an earlier version of the manuscript. We apologize for this error and it has now been corrected. Despite best intentions, errors do happen and this case illustrates why such meticulous reviewers are needed to ensure that we can all achieve the best research possible.

		before submission to academic journals. They are certainly able to do so as at least one of them has already published high quality inspiring research in the past. I am looking forward to reading more of that in the future. If you need any help or advice you know how to reach me. I am sorry I cannot be of more help at this point.	
--	--	--	--